# Enhancing Deep Batch Active Learning for Regression with Imperfect Data Guided Selection

**Yinjie Min***
School of Statistics and Data Science
Nankai University
nk.yjmin@gmail.com

**Furong Xu***
School of Statistics and Data Science
Nankai University
frxu@mail.nankai.edu.cn

**Xinyao Li**
School of Computer Science and Engineering
University of Electronic Science and Technology of China
xinyao326@std.uestc.edu.cn

**Changliang Zou**†
School of Statistics and Data Science, LPMC and KLMDASR and LEBPS
Nankai University
nk.chlzou@gmail.com

**Yongdao Zhou**†
School of Statistics and Data Science
Nankai University
ydzhou@nankai.edu.cn

## Abstract

Active learning (AL) reduces annotation costs by selecting the most informative samples based on both model sensitivity and predictive uncertainty. While sensitivity can be measured through parameter gradients in an unsupervised manner, predictive uncertainty can hardly be estimated without true labels especially for regression tasks, reducing the informativeness of actively selected samples. This paper proposes the concept of *auxiliary data* to aid the uncertainty estimation for regression tasks. With detailed theoretical analysis, we reveal that auxiliary data, despite potential distribution shifts, can provide a promising uncertainty surrogate when properly weighted. Such finding inspires our design of AGBAL, a novel AL framework that recalibrates auxiliary data losses through density ratio weighting to obtain reliable uncertainty estimates for sample selection. Extensive experiments show that AGBAL consistently outperforms existing approaches without auxiliary data across diverse synthetic and real-world datasets.

## 1 Introduction

Supervised machine learning often requires large amounts of labeled data to achieve good performance, but obtaining high-quality labels can be prohibitively expensive or time-consuming in many real-world applications[Sun et al., 2020a, Panayides et al., 2020, Thompson et al., 2024]. As a more

---

*Equal contribution
†Corresponding author

39th Conference on Neural Information Processing Systems (NeurIPS 2025).

efficient alternative, active learning (AL) strategically queries the most informative samples from the *target* unlabeled data pool to maximize model performance with minimal annotation effort[Gal and Ghahramani, 2016, Sener and Savarese, 2018, Beluch et al., 2018, Holzmüller et al., 2023].

Active learning aims to select the most informative samples for annotation. With the objective of minimizing the model's generalization loss through gradient-based parameter optimization, we establish that informativeness consists of two fundamental components: model sensitivity and predictive uncertainty. Highly sensitive samples produce stronger gradients during training, leading to more significant parameter updates[Koh and Liang, 2017, Chen et al., 2022], while those with high predictive uncertainty indicate regions where the predictions of the trained model are less accurate[Gal and Ghahramani, 2016, Lakshminarayanan et al., 2017]. By jointly considering both factors, we achieve more efficient model optimization through targeted sample selection.

The model sensitivity can be directly quantified using model gradients [Cai et al., 2013, Pinsler et al., 2019, Holzmüller et al., 2023]. For classification tasks, predictive uncertainty can be intuitively represented by certain metrics of class probabilities, such as entropy [Kirsch et al., 2019, Ash et al., 2020, Bang et al., 2024]. However, for regression tasks, the absence of labeled target data makes uncertainty estimation impossible, particularly since the model and training samples are not independent, and the loss on training data may approach zero during optimization. Various approaches have been proposed to find an alternative of predictive uncertainty in regression, including committee-based methods[Jose et al., 2024] and deep network dropout techniques[Gal and Ghahramani, 2016]. However, these solutions remain computationally intensive while offering only heuristic proxies rather than theoretically grounded measures. To address this, we propose using *auxiliary data* for robust uncertainty estimation and selecting samples with high informativeness.

We formalize *auxiliary data* as distributionally shifted yet relevant supplements to the target data in active learning, where the auxiliary distribution is more continuous than the target distribution. While target datasets are often limited by annotation costs, e.g. expert-labeled medical data, safety-critical driving scenarios or precise industrial design data records, auxiliary data naturally abound: hospitals accumulate medical images from patients with varying symptom manifestations[Litjens et al., 2017, Zhou et al., 2021, Tsai et al., 2024], autonomous vehicles continuously log driving data in varied environments[Sun et al., 2020b, Yu et al., 2020, Bai et al., 2024], and industrial systems retain sensor logs with recording inaccuracies or hardware degradation[Qiao et al., 2018, Rezazadeh et al., 2024]. Crucially, these auxiliary data share underlying physical or statistical relationships with the target distribution[Ganin et al., 2016, Kang et al., 2019, Zhang et al., 2020]. However, these imperfect datasets exhibit inherent limitations that constrain their utility for model training. Specifically, distributional shifts undermine the theoretical foundations of conventional generalization frameworks (e.g., Empirical Risk Minimization), leading to significant performance degradation[Buolamwini and Gebru, 2018, Alcorn et al., 2019, Koh et al., 2021]. Moreover, data contamination poses security threats, potentially enabling backdoor attacks and compromising data privacy in machine learning [Jagielski et al., 2018, Li et al., 2021, Carlini et al., 2021]. Thus, these data are frequently neglected by current AL methods.

Our key insight is that while auxiliary data cannot be directly combined with target data to improve model training performance, it can nevertheless provide reliable predictive uncertainty estimation through density ratio-weighted loss approximation, as shown in Figure 1. The proposed method can be briefly summarized in three steps: (1) estimating the density ratio between auxiliary and target distributions, (2) computing the model's loss on auxiliary data, and (3) applying density ratio weighting to obtain an approximation of the true loss. While the density ratio estimation is still dependent of the trained model, it successfully addresses the zero-loss problem and our analysis reveal that it yields a reliable surrogate for the true loss.

Inspired by such findings, we propose Auxiliary data Guided Batch Active Learning (AGBAL) to complement current AL selection frameworks. The contributions of this paper are as follows:

- We formally decompose informativeness in active learning into model sensitivity and predictive uncertainty for gradient-based parameter optimization. While the latter cannot be directly estimated due to model-training data dependence, we find that typically discarded imperfect auxiliary data can provide an uncertainty estimate surrogate when properly weighted.

- We propose Auxiliary data Guided Batch Active Learning (AGBAL), a novel framework that uses imperfect auxiliary data to estimate predictive uncertainty and selects more informative data. We validate the predictive uncertainty estimation through Neural Tangent Kernel (NTK) theory.

- Through extensive experiments on synthetic and real-world datasets, we demonstrate consistent performance advantages of AGBAL. Our comparative analysis reveals that auxiliary data-guided gradient kernels consistently outperform gradient kernels across various selection strategies.

## 2 Proposed Method

### 2.1 Problem Formulation

We consider multivariate regression and the goal is to learn a function $f : \mathbb{R}^d \to \mathbb{R}$ from training dataset $\mathcal{D}$, which consists of samples drawn i.i.d. from $P = P_X \times P_{Y|X}$. Typically, we consider a parameterized function family $\mathcal{F}(\Theta) = \{f(\cdot; \theta) : \theta \in \Theta\}, \Theta \subset \mathbb{R}^m$ and the goal is to find $\theta^* = \arg\min_{\theta \in \Theta} R(\theta; P)$, where $R(\theta; P) = \mathbb{E}_{X,Y \sim P} l(Y, f(X; \theta))$, and $l(\cdot, \cdot)$ is a loss function. In practice, we optimize $\theta$ by minimizing $\widehat{R}(\theta; \mathcal{D}) = |\mathcal{D}|^{-1} \sum_{(x,y) \in \mathcal{D}} l(y, f(x; \theta))$.

The goal of active learning (AL) is to select which data should be annotated in order to learn the model as quickly as possible. Given a training task objective $R(\theta; P)$, a batch mode active learning (BMAL) algorithm starts with a data pool $\mathcal{D}_X$, which consists of samples drawn i.i.d. from $P_X$. The initial set to be labeled is $\mathcal{B}_{0,X} \subset \mathcal{D}_X$, initial labeled set is $\mathcal{L}_0 = \mathcal{B}_0 = \{(x_i, y_i) \mid x_i \in \mathcal{B}_{0,X}, y_i \sim P_{Y|X=x_i}\}$ and the unlabeled set is $\mathcal{U}_0 = \mathcal{D}_X \setminus \mathcal{B}_{0,X}$.

Assume $\widehat{\theta}_{-1}$ is initialized randomly. At each BMAL step $t \geq 0$ we update the predictor parameter $\widehat{\theta}_t$ using $\mathcal{A}(\widehat{\theta}_{t-1}, \mathcal{L}_t)$, where $\mathcal{A} : \Theta \times (\mathcal{X} \times \mathcal{Y})^{\mathbb{N}} \to \Theta$ is the learning algorithm mapping previous parameter $\widehat{\theta}_{t-1}$ and labeled data $\mathcal{L}_t$ to an updated parameter $\widehat{\theta}_t$. Also assumes a next batch selection algorithm $\mathcal{S} : \Theta \times (\mathcal{X} \times \mathcal{Y})^{\mathbb{N}} \times \mathcal{X}^{\mathbb{N}} \times \mathbb{N} \times \mathcal{K} \to \mathcal{X}^{\mathbb{N}}$ that selects a batch of unlabeled data by $\mathcal{B}_{t+1,X} = \mathcal{S}(\widehat{\theta}_t, \mathcal{L}_t, \mathcal{U}_t, N, K)$, where $K(\cdot, \cdot) \in \mathcal{K}$ is a kernel that measures similarities between inputs. Label $\mathcal{B}_{t+1,X}$ and we get labeled dataset $\mathcal{B}_{t+1} = \{(x_i, y_i) \mid x_i \in \mathcal{B}_{t+1,X}, y_i \sim P_{Y|X=x_i}\}$. We update data pool by $\mathcal{L}_{t+1} = \mathcal{L}_t \cup \mathcal{B}_{t+1}, \mathcal{U}_{t+1} = \mathcal{U}_t \setminus \mathcal{B}_{t+1,X}$.

### 2.2 Motivation and Methods

Holzmüller et al. [2023] has demonstrated that sample selection strategies can be decomposed into two components: (1) a feature mapping, which projects input data into an informative representation space; and (2) a selection method operating in this space to identify samples of maximal value. Typically, the selection methods are implemented based on a distance metric in the representation space. In practice, this distance can be directly characterized by a kernel function defined over pairs of inputs. For example, the gradient kernel is defined as $K_{\mathrm{grad}}(x, x'; \theta) = \{\phi_1(\theta; x')\}^\top \phi_1(\theta; x)$, where $\phi_1(\theta; x) = \partial f(x; \theta)/\partial \theta \in \mathbb{R}^m$ is the model parameter gradient. Also a kernel function consists of a base kernel and a kernel transformation, where base kernel (like gradient kernel) serves for feature extraction and optional kernel transformations in section A.2 enable additional adaptation. For notation simplicity, kernel transformations are omitted in the main-text notation. Notably, the gradient kernel is essential for active learning frameworks by encoding the informativeness through parameter model gradients. While existing work Holzmüller et al. [2023] has demonstrated its superiority over alternatives like last-layer activations or NNGP kernels, we identify key limitations in its current formulation under regression tasks. The gradient of $R(\theta; P)$ at $\theta$ can be formulated as

$$\frac{\partial R(\theta; P)}{\partial \theta} = \mathbb{E}_{X,Y \sim P} \frac{\partial l(Y, f(X; \theta))}{\partial \theta} = \mathbb{E}_{X,Y \sim P} \frac{\partial l(Y, f(X; \theta))}{\partial f(X; \theta)} \frac{\partial f(X; \theta)}{\partial \theta}$$

$$= \mathbb{E}_{X \sim P_X} \phi_1(\theta; X) \left\{ \mathbb{E}_{Y \sim P_{Y|X}} \frac{\partial l(Y, f(X; \theta))}{\partial f(X; \theta)} \right\} = \mathbb{E}_{X \sim P_X} \phi_1(\theta; X) \phi_2(\theta; X), \quad (1)$$

where $\phi_2(\theta; x) = \mathbb{E}_{Y \sim P_{Y|X=x}} \partial l(Y, f(x; \theta))/\partial f(x; \theta)$ is the expected loss gradient. Given $\theta$ and input $x$, $\phi_1(\theta; x)$ can be directly computed, but evaluating $\phi_2(\theta; x)$ requires access to the conditional distribution $P_{Y|X=x}$. In an active learning setting, we face two fundamental challenges: (1) collecting additional samples from $P$ to estimate $\phi_2(\theta; x)$ is impractical, and (2) reusing existing training data introduces the double-dipping problem. Current methods[Holzmüller et al., 2023] directly set $\phi_2 \equiv 1$.

The gradient kernel relies solely on $\phi_1(\theta; x)$ to characterize the informativeness at point $x$. However, $\phi_2(\theta; x)$ captures the predictive uncertainty. When $f(x; \theta) = \arg\min_y \mathbb{E}_{Y \sim P_{Y|X=x}} l(Y, y)$, acquir-

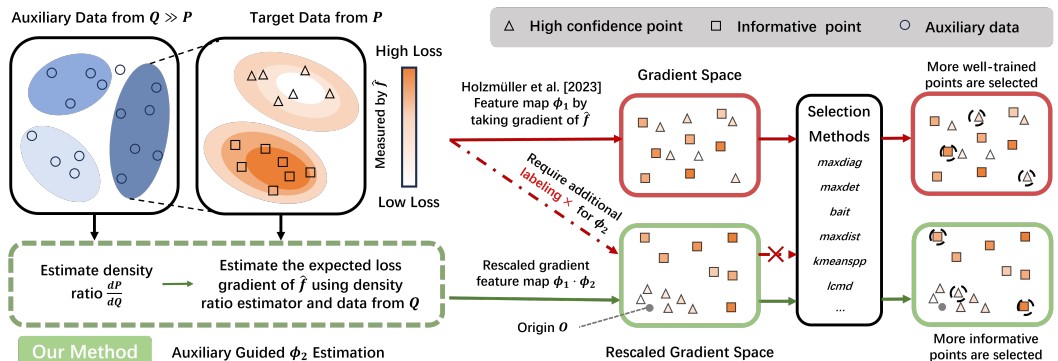

Figure 1: The AGBAL framework processes target data from $P$ and auxiliary data from $Q$. The rescaled gradient feature map projects high-confidence points near the origin $O$ (as their magnitudes of expected loss gradient are small), while maintaining high-uncertainty points at greater distances. Crucially, conventional methods without auxiliary data cannot estimate the expected loss gradient, causing high and low uncertainty points to distribute chaotically in the gradient space, hindering the selection of the most uncertain points.

ing additional samples at $x$ is meaningless. As shown in Figure 1 in the red box, many well-trained points can be selected. In practical scenarios, active learning task rarely operates in isolation: there often exists related tasks that possess *auxiliary dataset* that has underlying similarities but exhibit different data distribution. This motivates our **A**uxiliary data **G**uided **B**atch **A**ctive **L**earning (AG-BAL) framework in Figure 1: suppose we have access to a labeled dataset $\mathcal{D}' = \{(X_i', Y_i')\}_{i=1}^{n'}$ drawn from a different distribution $Q = Q_X \times Q_{Y|X} \neq P$, but $Q \gg P$ is absolutely continuous.

Let $g(x, y)$ and $g_{\mathrm{aux}}(x, y)$ denote the joint density functions of $P$ and $Q$, and $g(x), g_{\mathrm{aux}}(x)$ be their marginal density covariate functions. The conditional density is defined as $g(y \mid x) = g(x, y)/g(x)$ and $g_{\mathrm{aux}}(y \mid x) = g_{\mathrm{aux}}(x, y)/g_{\mathrm{aux}}(x)$. Define $r(x, y) = g(x, y)/g_{\mathrm{aux}}(x, y)$, $r(x) = g(x)/g_{\mathrm{aux}}(x)$ and $r(y \mid x) = r(x, y)/r(x)$. We can reformulate $\phi_2(\theta; x)$ as follows:

$$
\begin{aligned}
\phi_2(\theta; x) &= \mathbb{E}_{Y \sim P_{Y|X=x}}\left\{\frac{\partial l(Y, f(x; \theta))}{\partial f(x; \theta)}\right\} = \int g(y \mid x)\frac{\partial l(y, f(x; \theta))}{\partial f(x; \theta)}dy \\
&= \int g_{\mathrm{aux}}(y \mid x)r(y \mid x)\frac{\partial l(y, f(x; \theta))}{\partial f(x; \theta)}dy = \mathbb{E}_{Y \sim Q_{Y|X=x}}\left\{r(Y \mid x)\frac{\partial l(Y, f(x; \theta))}{\partial f(x; \theta)}\right\} \\
&= \arg\min_{s \in \mathbb{R}} \mathbb{E}_{Y \sim Q_{Y|X=x}}\left[\left\{s - \frac{\partial l(Y, f(x; \theta))}{\partial f(x; \theta)}\right\}^2 r(Y \mid x)\right]r(x) \\
&= \arg\min_{s \in \mathbb{R}} \mathbb{E}_{Y \sim Q_{Y|X=x}}\left[\left\{s - \frac{\partial l(Y, f(x; \theta))}{\partial f(x; \theta)}\right\}^2 r(x, Y)\right].
\end{aligned}
$$

Assume $\widehat{r}(x, y)$ is the density ratio estimator of $r(x, y)$ obtained through algorithm $\mathcal{A}_{\mathrm{dr}}$, which is detailed in section A.1, we can estimate $\phi_2$ in function space $\Phi$ by:

$$
\widehat{\phi}_2 = \arg\min_{\phi \in \Phi} \sum_{i=1}^{n'} \widehat{r}(X_i', Y_i')\left\{\phi(\theta; X_i') - \frac{\partial l(Y_i', f(X_i'; \theta))}{\partial f(X_i'; \theta)}\right\}^2. \tag{2}
$$

As shown in green dashed box in Figure 1, we define auxiliary data guided gradient feature map as $\phi_{\mathrm{aux}}(x; \theta; \phi_1, \widehat{\phi}_2) = \widehat{\phi}_2(\theta; x)\phi_1(\theta; x)$, and auxiliary data guided gradient kernel as

$$
K_{\mathrm{grad-aux}}(x, x'; \theta, \widehat{\phi}_2) = \{\phi_{\mathrm{aux}}(x'; \theta; \phi_1, \widehat{\phi}_2)\}^\top \phi_{\mathrm{aux}}(x; \theta; \phi_1, \widehat{\phi}_2). \tag{3}
$$

We present the complete algorithm for AGBAL in Algorithm 1.

## 2.3 Theoretical Analysis

In our proposed approach, the methodology originated from an error re-assessment framework designed to achieve a more precise feature characterization at each data point. We now proceed to

**Algorithm 1** Auxiliary Data Guided Batch Active Learning (AGBAL)

---

**Input:** Initial labeled and unlabeled set $\mathcal{L}_0$ and $\mathcal{U}_0$, auxiliary data set $\mathcal{D}'$, training algorithm $\mathcal{A}$, density ratio estimating algorithm $\mathcal{A}_{\mathrm{dr}}$, next batch selecting algorithm $\mathcal{S}$, loss function $l$, gradient mapping $\phi_1$, initial parameter $\widehat{\theta}_{-1}$, label budget $N$ at each step and training epochs $T$.

1: Set $t \leftarrow 0$
2: **While** $t < T$ **do**
3:     Update predictor parameter with $\widehat{\theta}_t \leftarrow \mathcal{A}(\widehat{\theta}_{t-1}, \mathcal{L}_t)$
4:     Estimate density ratio $\widehat{r}_t = \mathcal{A}_{\mathrm{dr}}(\mathcal{L}_t, \mathcal{D}')$
5:     Train estimator of expected loss gradient as $\widehat{\phi}_{2,t}$ in (2) using $\widehat{r}_t$ and $\mathcal{D}'$
6:     Construct auxiliary data guided gradient kernel $K_{t,\mathrm{grad-aux}}$ in (3) with $\widehat{\theta}_t$, $\phi_1$ and $\widehat{\phi}_{2,t}$
7:     Select unlabeled batch as $\mathcal{B}_{t,X} = \mathcal{S}(\widehat{\theta}_t, \mathcal{L}_t, \mathcal{U}_t, N, K_{t,\mathrm{grad-aux}})$
8:     Obtain $\mathcal{B}_t$ by labeling $\mathcal{B}_{t,X}$ and update $t \leftarrow t+1$, $\mathcal{L}_t \leftarrow \mathcal{L}_{t-1} \cup \mathcal{B}_t$, $\mathcal{U}_t \leftarrow \mathcal{U}_{t-1} \setminus \mathcal{B}_{t,X}$
9: **End while**
10: **return** $(\mathcal{L}_0, \ldots, \mathcal{L}_t)$ and $(\widehat{\theta}_0, \ldots, \widehat{\theta}_t)$

---

theoretically analyze the performance of auxiliary data based loss estimation under squared error loss $l_2(y_1, y_2) = (y_1 - y_2)^2/2$ in neural network architectures.

**Setting:** Consider data $(x_1, y_1), \ldots, (x_n, y_n)$ generated from model $y_i = f(x_i; \theta^*) + \epsilon_i$, where $\theta^* \in \Theta \subset \mathbb{R}^m$ represents the true parameters, and $\epsilon_i$ are independent noise terms with zero mean and finite variance $\sigma_\epsilon^2$. Define the empirical risk with a $L_2$ penalty as $\widehat{R}_2(\theta) = n^{-1} \sum_{i=1}^n l_2(y_i, f(x_i; \theta)) + \lambda \|\theta\|_2^2/2$, where $\lambda > 0$.

**Pointwise convergence of $f(\cdot; \widehat{\theta})$:** Let $H_{\mathrm{grad}}(\theta) \in \mathbb{R}^{n \times n}$ be the semidefinite matrix with $(i, j)$-entry $\{\phi_1(\theta; x_i)\}^\top \phi_1(\theta; x_j)$. The NTK kernel is defined as $K_{\mathrm{ntk}}(x_1, x_2) = \mathbb{E}_{\theta \sim P_\theta}[\{\phi_1(\theta; x_1)\}^\top \phi_1(\theta; x_2)]$. For $\mathbf{x} = (x_1, \ldots, x_n)$ and $\mathbf{y} = (y_1, \ldots, y_n)$, define $K_{\mathrm{ntk}}(x, \mathbf{x}) = (K_{\mathrm{ntk}}(x, x_1), \ldots, K_{\mathrm{ntk}}(x, x_n))$. As width goes to infinity and $\theta \sim P_\theta$, $H_{\mathrm{grad}}(\theta_0)$ converges in probability to a deterministic matrix $K_{\mathrm{ntk}}(\mathbf{x}, \mathbf{x}) \in \mathbb{R}^{n \times n}$ with entries $K_{\mathrm{ntk}}(x_i, x_j)$ [Jacot et al., 2018]. Denote $\widehat{f}_{\mathrm{ridge}}(x) = K_{\mathrm{ntk}}(x, \mathbf{x})\{K_{\mathrm{ntk}}(\mathbf{x}, \mathbf{x}) + \lambda I_n\}^{-1} \mathbf{y}^\top$, under certain conditions we have $f(x_0; \widehat{\theta})$ asymptotically equivalent to $\widehat{f}_{\mathrm{ridge}}(x_0)$ when the network width goes to infinity, and $\widehat{f}_{\mathrm{ridge}}(x_0)$ is asymptotically normal with $n \to \infty$. Details are presented in A.4 in supplementary materials.

**Re-formulation of $\widehat{\phi}_2$:** For $l_2$ loss $\partial l_2(y_1, y_2)/\partial y_2 = y_2 - y_1$. Therefore with $Y = f(X; \theta^*) + \epsilon$, we can calculate $\phi_2(\theta; x) = \mathbb{E}[f(X; \theta) - f(X; \theta^*) - \epsilon \mid X = x] = f(x; \theta) - f(x; \theta^*)$. Let $\phi_2(\widehat{\theta}; x_0) = \phi_2^{(1)}(\widehat{\theta}; x_0) + \phi_2^{(2)}(\widehat{\theta}; x_0)$, where $\phi_2^{(1)}(\widehat{\theta}; x_0) = \{f(x_0; \widehat{\theta}) - \widehat{f}_{\mathrm{ridge}}(x_0)\} + [\mathbb{E}\{\widehat{f}_{\mathrm{ridge}}(x_0)\} - f(x; \theta^*)]$ and $\phi_2^{(2)}(\widehat{\theta}; x_0) = [\widehat{f}_{\mathrm{ridge}}(x_0) - \mathbb{E}\{\widehat{f}_{\mathrm{ridge}}(x_0)\}]$. Assume $\widehat{r}$ is independent of $X_i', Y_i' \sim Q$, let distribution $Q_{\widehat{r}}$ satisfies $dQ_{\widehat{r}}(x, y)/dQ(x, y) = \widehat{r}(x, y)$. When $n' \to \infty$, the (2) can be re-formulated as

$$
\begin{aligned}
\widehat{\phi}_2 &= \arg\min_{\phi \in \Phi} \mathbb{E}_{x', y' \sim Q} \widehat{r}(x', y') \{\phi(\theta; x') - f(x'; \theta) + f(x'; \theta^*)\}^2 \\
&= \arg\min_{\phi \in \Phi} \mathbb{E}_{x, y \sim Q_{\widehat{r}}} [f(x; \theta^*) - \{f(x; \theta) - \phi(\theta; x)\}]^2 .
\end{aligned}
$$

Thus $\widetilde{f}(x_0; \widehat{\theta}) = f(x_0; \widehat{\theta}) - \widehat{\phi}_2(\widehat{\theta}; x_0)$ is an estimate of $f(x_0; \theta^*)$ and denote the corresponding asymptotic equivalent ridge kernel estimator as $\widetilde{f}_{\mathrm{ridge}}(x_0)$. Similarly define $\widetilde{\phi}_2(\widehat{\theta}; x_0) = \widetilde{f}(x_0; \widehat{\theta}) - f(x; \theta^*)$, $\widetilde{\phi}_2^{(1)}(\widehat{\theta}; x_0) = \{\widetilde{f}(x_0; \widehat{\theta}) - \widetilde{f}_{\mathrm{ridge}}(x_0)\} + [\mathbb{E}\{\widetilde{f}_{\mathrm{ridge}}(x_0)\} - f(x; \theta^*)]$ and $\widetilde{\phi}_2^{(2)}(\widehat{\theta}; x_0) = \widetilde{f}_{\mathrm{ridge}}(x_0) - \mathbb{E}\{\widetilde{f}_{\mathrm{ridge}}(x_0)\}$.

**Theorem 2.1.** *Assume equivalence $f(x_0; \widehat{\theta}) = \widehat{f}_{\mathrm{ridge}}(x_0) + o_P(\zeta(m))$, $\widetilde{f}(x_0; \widehat{\theta}) = \widetilde{f}_{\mathrm{ridge}}(x_0) + o_P(\zeta(m))$, where $m$ represents the neural network width and $\zeta(m) \to 0$ when $m \to \infty$, as well as asymptotic normality conditions $\phi_2^{(2)}(\widehat{\theta}; x_0) \xrightarrow{d} N(0, \sigma^2(x_0))$, $\widetilde{\phi}_2^{(2)}(\widehat{\theta}; x_0) \xrightarrow{d} N(0, \widetilde{\sigma}^2(x_0))$ and $\mathrm{Corr}(\phi_2^{(2)}(\widehat{\theta}; x_0), \widetilde{\phi}_2^{(2)}(\widehat{\theta}; x_0)) \to \rho$. We have*

$$
\mathrm{Var}(\widehat{\phi}_2(\widehat{\theta}; x_0)) \to \sigma^2(x_0) + \widetilde{\sigma}^2(x_0) - 2\rho \widetilde{\sigma}(x_0) \sigma(x_0) + o_P(\zeta(m)) .
$$

**Remark 2.2.** *As $\widehat{\phi}_2(\widehat{\theta}; x_0)$ is estimated on the holdout auxiliary dataset independent of $f(x_0; \widehat{\theta})$ and the only dependence is the density ratio estimator, we expect $\rho$ to be small. Also $\widetilde{f}(x_0; \widehat{\theta})$ is estimated similar to the process of the estimation of $f(x_0; \widehat{\theta})$, thus we expect $\sigma(x_0) \approx \widetilde{\sigma}(x_0)$. Therefore, $\widehat{\phi}_2(\widehat{\theta}; x_0)$ has variance approximate $2\sigma^2(x_0)$, which is proportion to the variance of $\widehat{f}_{\mathrm{ridge}}(x_0)$. Therefore $\widehat{\phi}_2(\widehat{\theta}; x_0)$ serves as a meaningful surrogate for the expected loss gradient.*

**Remark 2.3.** *In $\phi_2^{(1)}(\widehat{\theta}; x_0)$, the first term is negligible according to Lemma A.1 and the second term represents the bias of ridge estimation, which is not influenced by sampling randomness of $Y$ given $X = x_0$ and cannot be eliminated with limited samples under a given NTK kernel. The $\phi_2^{(2)}(\widehat{\theta}; x_0)$ is the variance component that characterizes the deviation from the expected estimation due to the training of finite samples. At locations where this term is large, additional sampling can improve the accuracy of the estimate. Although $\phi_2^{(2)}(\widehat{\theta}; x_0)$ cannot be directly calculated from $\widehat{f}_{\mathrm{ridge}}(\cdot)$, $\widehat{\phi}_2(\widehat{\theta}; x_0)$ can be calculated.*

## 3 Experiments

**Comparison Methods:** To evaluate our auxiliary data guided gradient kernel against the original gradient kernel framework, we conduct comprehensive comparisons across seven representative selection methods: (1) maxdiag (maximum diagonal selection) as the uncertainty-based baseline, (2) maxdet (determinant maximization) [Seo et al., 2000], (3) bait-FB (Forward-Backward-Greedy total uncertainty minimization) [Ash et al., 2020], (4) Frank-Wolfe (fw) optimization for kernel embedding approximation [Pinsler et al., 2019], (5) maxdist (maximum distance) representing geometric diversity approaches [Yu and Kim, 2010], (6) kmeanspp (next point probability proportional to squared distance) [Arthur and Vassilvitskii, 2006, Ostrovsky et al., 2013] and (7) lcmd (largest cluster maximum distance method) [Holzmüller et al., 2023], and (8) random selection. This systematic evaluation covers the spectrum from simple uncertainty sampling (maxdiag) to hybrid diversity/uncertainty methods (lcmd), allowing us to assess how our kernel optimization affects different selection paradigms. The codes are available in the repository `https://github.com/OswinMin/AGBAL`.

To ensure fair comparison, we standardize: (1) the neural network architecture across all methods, (2) initialization using parameters from identical $\mathcal{L}_0$ pretraining, (3) uniform hyperparameters (learning rates, epoch counts) during model updates, and (4) both kernel configurations ($K_{\mathrm{grad-aux}}$ and $K_{\mathrm{grad}}$) adopt the identical optimal transformation settings as established in Holzmüller et al. [2023].

All methods start with $|\mathcal{L}_0| = 200$ initially labeled samples and select $N = 200$ additional samples per step, running for $T = 15$ steps with an unlabeled pool size of $|\mathcal{U}_0| = 9,000$. The auxiliary dataset has size $N_{\mathrm{aux}} = 1000$.

**Evaluation Metrics:** In each experiment, the objective of optimization is the loss $l_2(y_1, y_2) = (y_1 - y_2)^2$, and the test set $\mathcal{T} = \{(\widetilde{x}_i, \widetilde{y}_i)\}_{i=1}^{|\mathcal{T}|}$ for evaluating the learner is distributed independently and identically (i.i.d.) with respect to the training data, with size $|\mathcal{T}| = 2000$. The mean squared error (MSE) of learner $f(\cdot; \theta)$ is defined as: $\mathrm{MSE} = \sum_{i=1}^{|\mathcal{T}|} l_2(\widetilde{y}_i, f(\widetilde{x}_i; \theta))$. For each data setting, every method is repeated 20 times.

- Let $\xi_0, \ldots, \xi_T$ denote the sequence of average MSE values for a given method; the area under the curve (AUC) of this sequence is calculated as: $\mathrm{AUC} = \sum_{i=1}^{T} (\xi_{i-1} + \xi_i)/2$.

- We report the average MSE for each method at specific training steps (i.e. the step 10), illustrating the training efficiency of different approaches under a fixed budget of training samples.

- Since AGBAL utilizes auxiliary data, we must verify that these data cannot be directly used for target task training. We compare the MSE of two models: one trained solely on $T$-step annotated samples, and another trained on both annotated samples and auxiliary data.

**Summary of Results:** We evaluate AGBAL (our method based on auxiliary data guided gradient kernel) against BMDAL (gradient kernel method without auxiliary data) and the random baseline across seven datasets: synthetic data S1, S2 and real-world datasets BIO, BIKE, DIAMOND, CT, STOCK. Details are in Section 3.1 and 3.2.

Table 1: Comparison of 8 selection methods across synthetic and real-world datasets in terms of AUC, where Avg Impro represents improvement over BMDAL averaged across 7 experiments.

| | S1 | S2 | BIO | BIKE | DIAMOND | CT | STOCK | Avg Impro |
|---|---|---|---|---|---|---|---|---|
| random | 0.928 | 1.421 | 0.451 | 0.459 | 21.009 | 0.380 | 0.392 | |
| lcmd | 1.011 | 1.517 | 0.417 | 0.394 | 19.714 | 0.255 | 0.370 | |
| lcmd (ours) | 0.846 | 1.279 | 0.420 | 0.435 | 20.687 | 0.291 | 0.363 | 0.6% |
| maxdist | 0.863 | 1.310 | 0.428 | 0.439 | 20.643 | 0.270 | 0.389 | |
| maxdist (ours) | 0.834 | 1.266 | 0.417 | 0.401 | 21.179 | 0.298 | 0.361 | 1.8% |
| kmeanspp | 0.894 | 1.378 | 0.414 | 0.404 | 19.691 | 0.271 | 0.372 | |
| kmeanspp (ours) | 0.842 | 1.294 | **0.406**$^*$ | 0.364 | **19.613**$^*$ | 0.264 | **0.355**$^*$ | 4.5% |
| fw | 0.953 | 1.448 | 0.434 | 0.455 | 21.115 | 0.347 | 0.388 | |
| fw (ours) | 0.899 | 1.341 | 0.418 | 0.404 | 21.840 | 0.310 | 0.377 | 5.5% |
| bait | 0.853 | 1.340 | 0.441 | 0.481 | 22.057 | 0.435 | 0.392 | |
| bait (ours) | 0.835 | 1.293 | 0.431 | 0.398 | 22.134 | 0.374 | 0.371 | 6.3% |
| maxdet | 0.876 | 1.318 | 0.418 | 0.486 | 19.702 | 0.320 | 0.378 | |
| maxdet (ours) | **0.833**$^*$ | **1.254**$^*$ | 0.409 | **0.362**$^*$ | 19.846 | **0.254**$^*$ | 0.359 | 8.9% |
| maxdiag | 0.903 | 1.401 | 0.451 | 0.597 | 24.594 | 0.526 | 0.415 | |
| maxdiag (ours) | 0.836 | 1.270 | 0.420 | 0.410 | 20.745 | 0.304 | 0.361 | 18.0% |

Table 1 reports the AUC of the learning curves, where lower values indicate faster convergence. AG-BAL consistently outperforms BMDAL in most settings, particularly with maxdet (achieving the best results on S1, S2, BIKE, and CT) and kmeanspp (leading on BIO, DIAMOND and STOCK). Table 2 compares the RMSE of AGBAL and BMDAL after 10 training steps across all selection strategies and datasets. AGBAL achieves significantly lower errors in almost all settings, demonstrating faster early-stage convergence.

Table 3 presents the test MSE comparison across all scenarios and methods after $T$-step active learning. To highlight the distributional discrepancy of auxiliary data, we deliberately select a large auxiliary dataset size ($N_{\text{aux}} = 10000$) to amplify the performance degradation caused by naively incorporating auxiliary data. The universally increased MSE values observed when incorporating auxiliary data into the training set demonstrates: (1) degraded model generalization performance, and (2) significant distributional shifts between auxiliary and target data. These results show that auxiliary data cannot be directly utilized for model training, thereby justifying our design of employing them exclusively for sample selection guidance.

Together, the empirical evidence establishes that while auxiliary data provide effective selection signals, their inherent distribution shifts prevent their direct inclusion in training sets.

We also investigate two other factors affecting AGBAL's performance: (1) auxiliary data quality (distributional discrepancy relative to the target distribution) in Section 3.1, and (2) auxiliary data quantity (sample size) under fixed quality conditions in Section A.7.2.

The random strategy, serving as a naive baseline, performs poorly, underscoring the importance of active selection. Notably, AGBALs gains are most pronounced for uncertainty-driven strategies (e.g., maxdet, kmeanspp), while methods perform less significant for hybrid and diversity-based selections like lcmd and maxdist. This demonstrates our methods compatibility with different selection criteria while maintaining superior convergence. For BMDAL without auxiliary data, kmeanspp and lcmd give the best performance.

## 3.1 Synthetic Data

In our synthetic experiments, the regression model for data from $P$ is $Y_i = \mu(X_i) + \epsilon(X_i)$, where $\mu(x) = \mathbb{E}(Y_i \mid X_i = x)$ and the residual $\epsilon(X_i)$ may depend on $X_i$. The model for data from $Q$ is $Y_i' = \mu'(X_i') + \epsilon'(X_i')$. The covariates $X_i$ and $X_i'$ are sampled from the $d$-dimensional standard normal distribution $N_d(\mathbf{0}_d, \mathbf{I}_d)$. We consider two data generating settings: for $x = (x_1, \ldots, x_d)$, let $\epsilon(x)$ and $\epsilon'(x) \sim \mathcal{N}(0, |\mu(x)|/4)$, $\mu'(x)$ is defined by shifting $\mu(x)$ with $\delta(x; \zeta) = \zeta \cdot \cos(\sum_{i=1}^{d} |x_i|)/4$:

**S1:** $\mu(x) = \sum_{i=1}^{d} |\log(|x_i| + 1)|/4$, and $\mu'(x) = \mu(x) + \delta(x; \zeta)$;

**S2:** $\mu(x) = \sum_{i=1}^{d} |x_i|^{1/2}/4$, and $\mu'(x) = \mu(x) + \delta(x; \zeta)$;

**Performance Result**: Figure 8 and 9 in appendix present the MSE learning curves versus the training steps for different sample selection methods with $\zeta = 8$. The results demonstrate that auxiliary data guided gradient kernel in AGBAL consistently outperforms the gradient kernel approach across nearly all experimental settings. Notably, while the gradient kernel may underperform random selection in certain cases (e.g., maxdiag on S1; maxdiag and fw on S2), AGBAL maintains superior performance over random selection in most of these scenarios. Refer to Table 4-5 for detailed MSE results.

**Experiments on auxiliary data quality**: We configure $\zeta \in \{2, 4, 8, 16, 32, 64\}$ to induce exponentially increasing distributional shifts in auxiliary data. As shown in Figure 2, AUC grows linearly with the exponential progression of $\zeta$. But our method still outperforms direct gradient kernel approaches within tested limits, as shown in Table 12. This demonstrates that: (1) higher-quality auxiliary data better guides active sample selection, and (2) even when auxiliary data quality deteriorates substantially, our loss estimation - serving as an error magnitude indicator rather than precise measurement - remains useful. More details are provided in Section A.7.1.

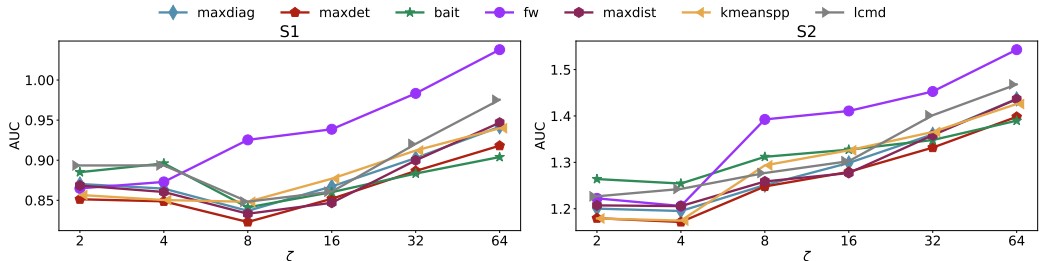

Figure 2: AUC plots of AGBAL under varying $\zeta$.

**Visual Analysis**: Further, we validate AGBAL's ability to identify sample points with higher predictive uncertainty through a controlled toy example. Using identical initially trained models and unlabeled data pools, we compare the predictive uncertainty of points selected via our auxiliary-guided gradient kernel versus conventional gradient kernels across different selection methods.

Let $\mu(x) = \sum_{i=1}^{d} x_i$, $\mu'(x) = 2\sum_{i=1}^{d} x_i$, and noise $\epsilon(x), \epsilon'(x) \sim \mathcal{N}(0, d)$. The initial labeled set $\mathcal{L}_0$ and the unlabeled pool $\mathcal{U}_0$ each contain 200 data points. In the subsequent batch selection step, we select 10 points from $\mathcal{U}_0$. For visualization, we project the unlabeled pool $\mathcal{U}_0$ onto a 2D space via PCA, with color mapping representing the expected loss gradient $f(x; \widehat{\theta}) - \mathbb{E}(Y \mid X = x)$. Darker shades indicate higher predictive uncertainty, which should be prioritized for selection. As illustrated in Figure 3, while both AGBAL and BMDAL aim to select representative points in the representation space, AGBAL consistently identifies more high-uncertainty samples. The result plots for other selection methods are provided in Figure 5-7.

## 3.2 Real Data

We evaluate our method on five public regression datasets also considered by [Holzmüller et al., 2023]: physicochemical properties of protein tertiary structure (BIO) [Rana, 2013], bike sharing (BIKE) [Fanaee-T and Gama, 2014], prices of diamonds (DIAMOND) , relative location of CT slices (CT) [Graf and Cavallaro, 2011], and BNG stock price data (STOCK) . Detailed dataset descriptions are provided in Section A.6.

**Result**: Figure 4 presents the MSE learning curves comparing different methods on dataset BIKE (with additional results on four other real-world datasets shown in Figures 10-13 and Table 6-9 in appendix due to space constraints). The results demonstrate that our auxiliary data guided gradient kernel based methods achieves the fastest MSE reduction compared with gradient kernel based methods, which indicates $K_{\text{grad-aux}}$ better measures the informativeness of data points. Moreover, while lcmd and kmeanspp typically perform best without auxiliary data, the maxdet and kmeanspp

---

On https://www.kaggle.com/datasets/resulcaliskan/diamonds
On https://www.openml.org/search?type=data&sort=runs&id=1200&status=active

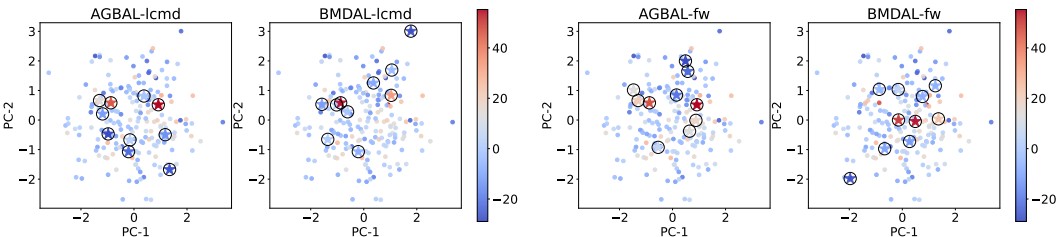

Figure 3: Visualization of the loss of selected points across four AL configurations. Left, right panels display lcmd, fw results of AGBAL and BMDAL, respectively.

selection methods show superior performance when guided by auxiliary data. This consistent pattern across all real-data scenarios confirms the effectiveness of incorporating auxiliary information into the sample selection process.

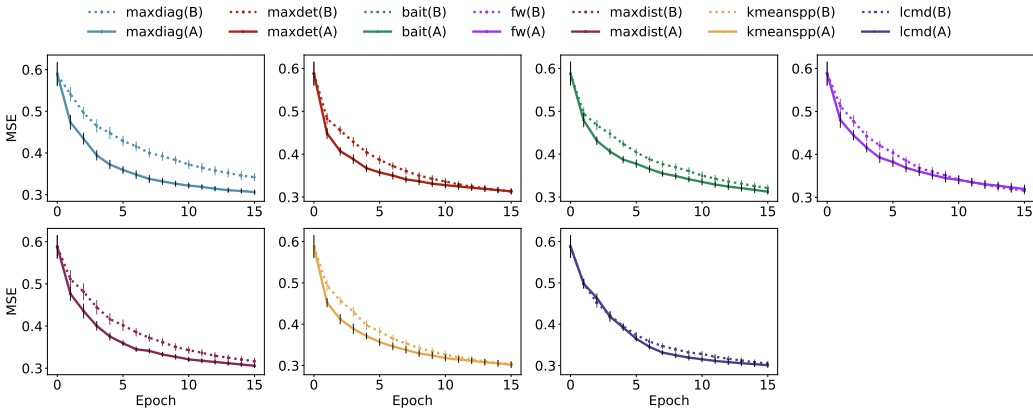

Figure 4: MSEs decreasing plots during AL steps for different selection method and different kernels for STOCK dataset (B for gradient kernel, A for auxiliary data guided gradient kernel)

### 3.3 Computational Burden Analysis

Our method introduces additional components (density ratio estimation and auxiliary loss estimation) compared to gradient-only approaches BMDAL. However the selection kernel itself remains unchanged; we only modify the features used for selection. The primary computational overhead stems from processing auxiliary data. For efficiency, lightweight machine learning methods (e.g., random forests) can be adopted for density ratio and loss estimation without compromising performance. To illustrate scalability, we conducted experiments across 5 real-world datasets with varying auxiliary dataset sizes (100, 500, 1000, 10,000 samples) on a server with dual Intel Xeon Gold 6330 CPUs (112 threads total) and 125GB RAM.

The results are shown in the Table 11. Even with only 100 auxiliary data, the maxdet selection strategy, which performs best among the compared approaches, still achieves meaningful performance improvements. The gain grows with larger auxiliary datasets. Crucially, for auxiliary datasets below 1,000 samples, the additional time and memory overhead remains negligible. In practical applications, we can choose the auxiliary data size based on their specific needs for either better performance (larger auxiliary sets) or faster computation (smaller auxiliary sets).

## 4 Related Work

In Section 1 and 2.2, we propose that in regression tasks under gradient-based optimization, informativeness can be decomposed into model sensitivity and predictive uncertaintytwo dimensions that

have been extensively studied separately. In this section, we systematically review existing work along these two research lines.

Expected model change maximization plays a vital role in model sensitivity-based methods. Early work by Cai et al. [2013] introduced bootstrap ensemble modeling to identify samples with maximal gradient variations across ensemble members, explicitly capturing model sensitivity. Subsequent approaches integrated representation learning: Ash et al. [2020] developed a clustering strategy using last-layer gradient embeddings to select samples that induce divergent model updates, while Holzmüller et al. [2023] proposed a gradient-based kernel transformation method that strategically selects the most distant points within the largest cluster, optimizing both representativeness and diversity.

As predictive uncertainty cannot be directly estimated, various alternative methods have been proposed. Early work includes RayChaudhuri and Hamey [1995] and Burbidge et al. [2007], who introduced the Query-by-Committee (QBC) framework to select samples with maximal model disagreement as a measure of uncertainty. With the advent of deep learning, more sophisticated uncertainty estimation techniques emerged. Gal and Ghahramani [2016] developed Monte Carlo dropout to approximate predictive uncertainty, while Beluch et al. [2018] employed ensembles with varied parameter initializations to characterize sample uncertainty.

Alternative approaches focus on diversity and representativeness. Sener and Savarese [2018] introduced the core-set approach with theoretical guarantees via minimum covering radius, and Wu et al. [2019] developed a greedy maxmin distance algorithm for data selection. Subsequent developments include Liu et al. [2021]'s iterative optimization method balancing average and minimum distances, and Kim and Shin [2022]'s hybrid approach combining density-based clustering and core-set selection to enhance diversity-representativeness trade-offs.

A distinct line of research explores loss-driven approaches. Konyushkova et al. [2017] proposed a data-driven method that learns query strategies by predicting expected error reduction. Yoo and Kweon [2019] learned a loss predictor module to estimate target losses of unlabeled data. Sinha et al. [2019] introduced VAAL, which uses adversarial training between a variational autoencoder and a discriminator to learn latent representations while implicitly quantifying sample loss.

Several works also leverage auxiliary data for active learning. Our approach differs from transfer active learning methods such as Wang et al. [2014] in how auxiliary data is defined: transfer learning and domain adaptation typically assume covariate shift with different $P(X)$ but similar $P(Y \mid X)$ across domains, whereas our setting allows both $P(X)$ and $P(Y \mid X)$ to differ in imperfect auxiliary data. This distinction is critical, as standard transfer methods may experience more negative transfer under such joint distribution shifts.

# 5   Conclusion

In this work, we address a fundamental challenge in active learning for regression tasks - the inability to reliably estimate predictive uncertainty. We propose AGBAL, a novel active learning framework that overcomes this limitation by leveraging auxiliary data through density ratio-weighted loss approximation. Theoretically grounded in NTK analysis, our method transforms typically discarded auxiliary data into valuable uncertainty estimates, while properly accounting for distributional shifts between auxiliary and target domains. Through extensive evaluations, we demonstrate that AGBAL consistently outperforms conventional active learning approaches across diverse application scenarios.

## Acknowledgments and Disclosure of Funding

Zou was supported by the National Key R&D Program of China (Grant No. 2022YFA1003800) and the National Natural Science Foundation of China (Grant Nos. 12231011 and 12531011).

Zhou was supported by the National Natural Science Foundation of China (Grant Nos. 12131001), and the Fundamental Research Funds for the Central Universities in Nankai University.

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

# A Technical Appendices and Supplementary Material

## A.1 Density Ratio Estimation

Consider two datasets $\mathcal{D}_P = \{x_i\}_{i=1}^n \sim P$ with density $p(x)$, and $\mathcal{D}_Q = \{z_j\}_{j=1}^m \sim Q$ with density $q(z)$. Assume $P \ll Q$, the target is to estimate the density ratio $r(x) = dP/dQ(x) = p(x)/q(x)$, which quantifying the discrepancy between the distribution $P$ and $Q$.

There are two kind of density ratio estimation algorithms $\mathcal{A}_{\mathrm{dr}}(\cdot)$: direct estimation and classifier-based indirect estimation.

In direct density ratio estimation, we typically use a parametric function $g_\theta(x)$ to estimate $r(x)$. The estimation problem is then formulated as minimizing a minimizing the Bregman divergence between $p(x)$ and $g_\theta(x)q(x)$: given a certain function $u(x)$,

$$\min_\theta \{\mathbb{E}_{x \sim Q}[\partial u(g_\theta(x))g_\theta(x) - u(g_\theta(x))] - \mathbb{E}_{x \sim P}[\partial u(g_\theta(x))]\}.$$

There are many choices for the function $u(x)$, such as: (1) $u(x) = (x-1)^2/2$ used by LSIF [Kanamori et al., 2009], (2) $u(x) = x \log(x) - x$ used by UKL [Nguyen et al., 2010], (3) $u(x) = x \log(x) - (1+x) \log(1+x)$ used by BKL (LR) [Hastie et al., 2005].

For undirected density ratio estimation, we adopt a probabilistic classification framework. We construct a labeled dataset by assigning label 1 to samples from $\mathcal{D}_P$ and label 0 to samples from $\mathcal{D}_Q$. Specifically, we first train a classifier to distinguish samples from the target versus mixture distributions, then apply calibration methods [Zadrozny and Elkan, 2002], i.e., Platt-calibration [Platt et al., 1999] to convert classifier outputs into well-calibrated probability estimates.

The combined dataset $\{(W_i, Y_i)\}_{i=1}^{n+m}$ contains $n + m$ samples, where the first $n$ observations are from $\mathcal{D}_P$, $(W_i, Y_i) = (x_i, 1), i \le n$ and $(W_i, Y_i) = (z_i, 0), i > n$. The conditional class probability can be expressed as:

$$\mathbb{P}(Y = 1 \mid W = w) = \frac{p(w)}{p(w) + q(w)} = \frac{r(w)}{r(w) + 1},$$

where $r(w) = p(w)/q(w)$ is the density ratio function. This relationship allows us to recover the density ratio through:

$$r(w) = \frac{\mathbb{P}(Y = 1 \mid W = w)}{1 - \mathbb{P}(Y = 1 \mid W = w)}.$$

Let $\widetilde{g}(x)$ be a probabilistic classifier trained to estimate $\mathbb{P}(Y = 1 \mid W = w)$. Assume calibrated classifier as $\widehat{g}(x) = \sigma(a\widetilde{g}(x) + b)$, where $\sigma$ is the sigmoid function and $a, b$ are trained on another dataset. In practice we can split $\{(W_i, Y_i)\}_{i=1}^{n+m}$ into two parts: one to train $\widetilde{g}(x)$ and another to fit $a$ and $b$. The corresponding density ratio estimator is then given by:

$$\widehat{r}(x) = \frac{\widehat{g}(x)}{1 - \widehat{g}(x)}.$$

## A.2 Kernel Transformations

The framework by [Holzmüller et al., 2023] introduces a modular set of kernel transformations to adapt base kernels for active learning objectives. These transformations modify base kernels to improve uncertainty estimation (e.g., Gaussian process posterior transformation), diversity (e.g., acs random feature transformation and acs gradient transformation), and computational efficiency (e.g., scaling, sketching). All transformations are composable and can be sequentially chained to combine their effects. We denote this transformation chain as $K_{\to T_1 \to T_2}$, where $T_1$ is first applied to the base kernel $K$, followed by $T_2$. In the following, we formalize each transformation and its implementation.

To simplify our notation, we consider a dataset $\mathcal{D}_X$, which is partitioned into two disjoint subsets: a labeled subset $\mathcal{L}_X = \{x_i\}_{i=1}^n \subset \mathcal{D}_X$ that has been annotated by domain experts, resulting in the labeled dataset $\mathcal{L} = \{(x_i, y_i)\}_{i=1}^n$. An unlabeled subset $\mathcal{U} = \mathcal{D}_X \setminus \mathcal{L}_X$ containing the remaining unannotated data points. We assume the existence of a positive semidefinite kernel function $K : \mathcal{X} \times \mathcal{X} \to \mathbb{R}$ defined in the input space.

(1) Scaling transformation $\rightarrow \text{scale}(\mathcal{L}_X)$ normalizes kernel outputs to have a unit variance:

$$K_{\rightarrow \text{scale}(\mathcal{L}_X)}(x, x') = \kappa^2(K, \mathcal{L}_X)K(x, x'), \quad \kappa(K, \mathcal{L}_X) = \left( \frac{1}{|\mathcal{L}_X|} \sum_{x \in \mathcal{L}_X} K(x, x) \right)^{-1/2},$$

(2) Sketching transformation $\rightarrow \text{sktch}(d_2)$ summarizes feature mapping with a projection to lower dimension space:

$$K_{\rightarrow \text{sktch}(d_2)}(x, x') = \frac{1}{p}\{\phi(x)\}^\top N^\top N \phi(x'),$$

where $\phi(x)$ is the map corresponding to the base kernel $K$ that $K(x, x') = \{\phi(x)\}^\top \phi(x')$, where $\phi(x) \in \mathbb{R}^{d_1}$ and $N \in \mathbb{R}^{d_2 \times d_1}, d_2 < d_1$ is a random matrix with $i.i.d.$ standard normal entries, approximates high-dimensional features via random projection to $d_2$ dimensions. Using variants of the celebrated Johnson-Lindenstrauss lemma[Johnson et al., 1984], we can prove that for a Gaussian random projection to dimension $d_2 \geq \log(d_1^2/\delta)/\epsilon^2$, with probability $\geq 1 - \delta$ we have:

$$\forall x, x' \in \mathcal{X} : (1 - \epsilon)d_K(x, x') \leq d_{K_{\rightarrow \text{sketch}(p)}}(x, x') \leq (1 + \epsilon)d_K(x, x'),$$

where $d_K(x, x') = \sqrt{K(x, x) + K(x', x') - 2K(x, x')}$ denotes the kernel distance between $x$ and $x'$.

(3) Gaussian process posterior transformation: For the feature map $\phi(x)$ corresponding to the kernel $K$, consider a Gaussian process with kernel $K$. Assume a Gaussian process linear regression model using $\phi(x)$ as features: $y_i = \omega^\top \phi(x_i) + \epsilon_i$ with a weight prior $\omega \sim \mathcal{N}(0, \mathbf{I})$ and i.i.d. observation noise $\epsilon_i \sim \mathcal{N}(0, \sigma^2)$. Bishop and Nasrabadi [2006] proves the posterior distribution for a Gaussian process after observing the training data $\mathcal{L}$ is also a Gaussian process with kernel

$$K_{\rightarrow \text{post}(\mathcal{L}, \sigma^2)}(x, x') = \text{Cov}\left( \omega^\top \phi(x), \omega^\top \phi(x') \mid \mathcal{L} \right)$$
$$= K(x, x') - K(x, \mathcal{L}_X)\left( K(\mathcal{L}_X, \mathcal{L}_X) + \sigma^2 I \right)^{-1} K(\mathcal{L}_X, x'),$$

where $K(x, \mathcal{L}_X) = \left( K(x, x_1), K(x, x_2), \cdots, K(x, x_n) \right)$, $K(\mathcal{L}_X, x) = \{K(x, \mathcal{L}_X)\}^\top$ and $K(\mathcal{L}_X, \mathcal{L}_X) = (K(x_i, x_j))_{i,j}$. Calculate the posterior covariance of the Gaussian process after observing data $\mathcal{L}$ with noise variance $\sigma^2$. For convenience, we use $K_{\rightarrow \mathcal{L}}$ to denote $K_{\rightarrow \text{scale}(\mathcal{L}) \rightarrow \text{post}(\mathcal{L}, \sigma^2)}$.

(4) ACS random feature transformation: Pinsler et al. [2019] applied the ACS-FW method to Gaussian process linear regression. We use the Gaussian process model parameterized by $\omega$ with noise variance $\sigma^2$, employing the kernel $K_{\rightarrow \text{scale}(\mathcal{L})}$ as described above. Let $f_{\text{acs}}(x, \omega) = 1/2 \log\{1 + K_{\rightarrow \mathcal{L}}(x, x)/\sigma^2\} - [\{\omega^\top \phi_{\rightarrow \text{scale}(\mathcal{L}_X)}(x)\}^2 + K_{\rightarrow \mathcal{L}}(x, x)]/2\sigma^2$, where $\phi_{\rightarrow \text{scale}(\mathcal{L}_X)}$ denotes the feature mapping associated with the kernel function $K_{\rightarrow \text{scale}(\mathcal{L}_X)}$. Then,

$$K_{\rightarrow \text{acs}}(x, x') = \mathbb{E}_{\omega \sim P(\omega | \mathcal{L})}\{f_{\text{acs}}(x, \omega) f_{\text{acs}}(x', \omega)\},$$

where $P(\omega \mid \mathcal{L})$ denotes the posterior distribution of parameter $\omega$ observing data $\mathcal{L}$.

(5) ACS gradient transformation: Pinsler et al. [2019] propose the weighted Fisher inner product given by:

$$K_{\rightarrow \text{acs-grad}}(x, x') = \mathbb{E}_{\omega \sim P(\omega | \mathcal{L})}[\{\nabla_\omega f_{\text{acs}}(x, \omega)\}^\top \nabla_\omega f_{\text{acs}}(x', \omega)].$$

Under the Gaussian process model, they show an explicit formula for $K_{\rightarrow \text{acs-grad}}(x, x')$ is given by:

$$K_{\rightarrow \text{acs-grad}}(x, x') = \frac{1}{\sigma^4} K_{\rightarrow \text{scale}(\mathcal{L})}(x, x') K_{\rightarrow \mathcal{L}}(x, x').$$

## A.3 Selection Methods

After applying the transformed base kernels to the unlabeled data, multiple selection strategies can be employed to identify the most informative unlabeled candidates for annotation.

Let $\mathcal{L}_t = \{(x_i, y_i)\}_{i=1}^n$ represent the labeled dataset where $\mathcal{L}_{t,X} = \{x_i\}_{i=1}^n$ contains only the input features, $\mathcal{U}_t = \{x_j\}_{j=1}^m$ denotes the unlabeled data pool, $\widehat{\theta}_t$ represents the current model

parameters, and $K$ denotes the kernel of the base kernel after kernel transformations. The active learning objective is to select $N$ points from $\mathcal{U}_t$ for annotation in the next batch.

We employ an iterative approach to select samples (except bait-FB). Let $\mathcal{B}_{t,X,i}$ denote the selected sample batch after the $i$-th iteration, where $|\mathcal{B}_{t,X,i}| = i$. By repeating the selection process $N$ times for different methods and combining the selected samples, we obtain the next batch of points for labeling. The next batch selection algorithm is denoted as $\mathcal{S} : \Theta \times (\mathcal{X} \times \mathcal{Y})^{\mathbb{N}} \times \mathcal{X}^{\mathbb{N}} \times \mathbb{N} \times \mathcal{K} \to \mathcal{X}^{\mathbb{N}}$. The next batch of unlabeled data is $\mathcal{B}_{t+1,X} = \mathcal{S}(\widehat{\theta}_t, \mathcal{L}_t, \mathcal{U}_t, N, K)$, where $N$ is the selected batch size. We denote the selection at the $i$-th iteration with $\mathcal{B}_{t,X,i}$ as $\mathcal{S}(\widehat{\theta}_t, \mathcal{L}_t, \mathcal{U}_t, \mathcal{B}_{t,X,i}, K)$

Maxdiag selection maximizes the diagonal entries of the kernel matrix to prioritize high-uncertainty samples:

$$\mathcal{S}(\widehat{\theta}_t, \mathcal{L}_t, \mathcal{U}_t, \mathcal{B}_{t,X,i}, K) = \operatorname*{argmax}_{x \in \mathcal{U}_t} K(x,x),$$

Maxdet [Seo et al., 2000] selection optimizes the determinant of the kernel matrix, balancing informativeness and diversity:

$$\mathcal{S}(\widehat{\theta}_t, \mathcal{L}_t, \mathcal{U}_t, \mathcal{B}_{t,X,i}, K) = \operatorname*{argmax}_{x \in \mathcal{U}_t} K_{\to \mathrm{post}(\mathcal{B}_{t,X,i}, \sigma^2)}(x,x),$$

where $\sigma^2$ is a tuning parameter.

Bait-FB [Ash et al., 2020] selection leverages Fisher information for batch selection to enhance representativeness. Bait-FB greedily selects $2N$ samples and then removes $N$ samples, and the greedy selection of $2N$ samples is:

$$\mathcal{S}(\widehat{\theta}_t, \mathcal{L}_t, \mathcal{U}_t, \mathcal{B}_{t,X,i}, K) = \operatorname*{argmax}_{x \in \mathcal{U}_t} \sum_{\tilde{x} \in \mathcal{L}_{t,X} \cup \mathcal{U}_t} K_{\to \mathrm{post}(\mathcal{B}_{t,X,i} \cup \{x\}, \sigma^2)}(\tilde{x}, \tilde{x}).$$

Frank-Wolfe selection employs a convex optimization approach to iteratively select batches with near-optimal submodular guarantees. Details can be referred to Pinsler et al. [2019].

Maxdist selects samples that maximize pairwise distances, and explicitly promotes diversity [Yu and Kim, 2010]:

$$\mathcal{S}(\widehat{\theta}_t, \mathcal{L}_t, \mathcal{U}_t, \mathcal{B}_{t,X,i}, K) = \operatorname*{argmax}_{x \in \mathcal{U}_t} \min_{x' \in \mathcal{B}_{t,X,i}} d_K(x,x').$$

For $\mathcal{B}_{t,X,0}$, an arbitrary maximizer from $\mathcal{U}_t$ is chosen.

Kmeanspp ensures representative coverage via k-means++ initialization[Arthur and Vassilvitskii, 2006, Ostrovsky et al., 2013]:

$$\forall x \in \mathcal{U}_t : P(\mathcal{S}(\widehat{\theta}_t, \mathcal{L}_t, \mathcal{U}_t, \mathcal{B}_{t,X,i}, K) = x) = \frac{\min_{\tilde{x} \in \mathcal{B}_{t,X,i}} d_K(x, \tilde{x})^2}{\sum_{x' \in \mathcal{U}_t} \min_{\tilde{x} \in \mathcal{B}_{t,X,i}} d_K(x', \tilde{x})^2}.$$

Lcmd Holzmüller et al. [2023] selects the point with the maximum distance to point in $\mathcal{B}_{t,X,i}$ that has largest cluster size. For each point $x \in \mathcal{U}_t$ define its associated center as $c(x) = \arg\min_{\tilde{x} \in \mathcal{B}_{t,X,i}} d_K(x, \tilde{x})$. For any $\tilde{x} \in \mathcal{B}_{t,X,i}$, define its cluster size as $s(\tilde{x}) = \sum_{x \in \mathcal{U}_t : c(x) = \tilde{x}} d_K(x, \tilde{x})^2$. The selection is defined as:

$$\mathcal{S}(\widehat{\theta}_t, \mathcal{L}_t, \mathcal{U}_t, 1, K) = \operatorname*{argmax}_{x \in \mathcal{U}_t : s(c(x)) = \max_{\tilde{x} \in \mathcal{B}_{t,X,i}} s(\tilde{x})} d_K(x, c(x)).$$

For $\mathcal{B}_{t,X,0}$, we select $\operatorname{argmax}_{x \in \mathcal{U}_t} K(x,x)$.

### A.4 Theoretical Details

With the NTK kernel defined as $K_{\mathrm{ntk}}(x_1, x_2) = \mathbb{E}_{\theta \sim P_\theta}\big[\{\phi_1(\theta; x_1)\}^\top \phi_1(\theta; x_2)\big]$, the key equivalence holds:

**Lemma A.1.** *[Yang, 2019, Arora et al., 2019] For data $(x_1, y_1), \ldots, (x_n, y_n)$, as layer widths go to infinity and the parameters of neural network are initialized by $\theta \sim P_\theta$, the optimized predictor $f(x; \widehat{\theta})$ under $\widehat{R}_2$ converges in probability to:*

$$\widehat{f}_{\mathrm{ridge}}(x) = K_{\mathrm{ntk}}(x, \mathbf{x})\big\{K_{\mathrm{ntk}}(\mathbf{x}, \mathbf{x}) + \lambda I_n\big\}^{-1} \mathbf{y}^\top.$$

Therefore, under the infinite-width regime, the equivalence between neural network training and kernel ridge regression allows us to leverage pointwise behavior of $\widehat{f}_{\mathrm{ridge}}(x)$ to characterize the neural network predictor at a fixed point $x_0 \in \mathbb{R}^d$. Denote ridge estimate of NTK kernel as:

$$\widehat{K}_{\mathrm{ntk}}(x, x_0) = K_{\mathrm{ntk}}(x, \mathbf{x})\big\{K_{\mathrm{ntk}}(\mathbf{x}, \mathbf{x}) + \lambda n I_n\big\}^{-1}\big\{K_{\mathrm{ntk}}(x_0, \mathbf{x})\big\}^\top.$$

Let $L_\infty$ norm be $\|\widehat{K}_{\mathrm{ntk}}(\cdot, x_0) - K_{\mathrm{ntk}}(\cdot, x_0)\|_\infty = \sup_x |\widehat{K}_{\mathrm{ntk}}(x, x_0) - K_{\mathrm{ntk}}(x, x_0)|$, and empirical norm be $\|\widehat{K}_{\mathrm{ntk}}(\cdot, x_0) - K_{\mathrm{ntk}}(\cdot, x_0)\|_n = \big[n^{-1}\sum_{i=1}^n \{\widehat{K}_{\mathrm{ntk}}(x_i, x_0) - K_{\mathrm{ntk}}(x_i, x_0)\}^2\big]^{1/2}$.

**Assumption 1.** The noise terms $\epsilon_i$ are independent and identically distributed with $\mathbb{E}(\epsilon_i) = 0$ and $\mathrm{Var}(\epsilon_i) = \sigma_\epsilon^2 < \infty$

**Assumption 2.** $n^{-1/2}\|\widehat{K}_{\mathrm{ntk}}(\cdot, x_0) - K_{\mathrm{ntk}}(\cdot, x_0)\|_\infty / \|\widehat{K}_{\mathrm{ntk}}(\cdot, x_0) - K_{\mathrm{ntk}}(\cdot, x_0)\|_n \to 0$ in probability.

**Lemma A.2.** *[Tuo and Zou, 2024] Under assumption 1-2, $\widehat{f}_{\mathrm{ridge}}(x_0)$ is asymptotically normal:*

$$\{\mathrm{Var}(\widehat{f}_{\mathrm{ridge}}(x_0))\}^{-1/2}\Big[\widehat{f}_{\mathrm{ridge}}(x_0) - \mathbb{E}\{\widehat{f}_{\mathrm{ridge}}(x_0)\}\Big] \xrightarrow{d} N(0, 1),$$

*where* $\mathrm{Var}(\widehat{f}_{\mathrm{ridge}}(x_0)) = \sigma_\epsilon^2 K_{\mathrm{ntk}}(x_0, \mathbf{x})\big\{K_{\mathrm{ntk}}(\mathbf{x}, \mathbf{x}) + \lambda n I_n\big\}^{-1}\big\{K_{\mathrm{ntk}}(x_0, \mathbf{x})\big\}^\top.$

**Remark A.3.** *Assumption 2 can be verified using the upper and lower bounds of $\|\widehat{K}_{\mathrm{ntk}}(\cdot, x_0) - K_{\mathrm{ntk}}(\cdot, x_0)\|_\infty$ and $\|\widehat{K}_{\mathrm{ntk}}(\cdot, x_0) - K_{\mathrm{ntk}}(\cdot, x_0)\|_n$ in Tuo and Zou [2024].*

## A.5   Proof of Theorem 2.1

We can decompose the $\widehat{\widetilde{\phi}}_2(\widehat{\theta}; x_0)$ as

$$
\begin{aligned}
\widehat{\widetilde{\phi}}_2(\widehat{\theta}; x_0) &= f(x_0; \widehat{\theta}) - \widetilde{f}(x_0; \widehat{\theta}) \\
&= \big\{f(x_0; \widehat{\theta}) - f(x_0; \theta^*)\big\} - \big\{\widetilde{f}(x_0; \widehat{\theta}) - f(x_0; \theta^*)\big\} \\
&= \phi_2(\widehat{\theta}; x_0) - \widetilde{\phi}_2(\widehat{\theta}; x_0) \\
&= \big\{\phi_2^{(1)}(\widehat{\theta}; x_0) - \widetilde{\phi}_2^{(1)}(\widehat{\theta}; x_0)\big\} + \big\{\phi_2^{(2)}(\widehat{\theta}; x_0) - \widetilde{\phi}_2^{(2)}(\widehat{\theta}; x_0)\big\} \\
&= \big[\mathbb{E}\{\widehat{f}_{\mathrm{ridge}}(x_0)\} - \mathbb{E}\{\widetilde{f}_{\mathrm{ridge}}(x_0)\} + o_P(\zeta(m))\big] + \big\{\phi_2^{(2)}(\widehat{\theta}; x_0) - \widetilde{\phi}_2^{(2)}(\widehat{\theta}; x_0)\big\}.
\end{aligned}
$$

Therefore the first part only contains bias and an asymptotically negligible item and the variance comes from the second part:

$$
\begin{aligned}
&\mathrm{Var}(\widehat{\widetilde{\phi}}_2(\widehat{\theta}; x_0)) \\
=\,&\mathrm{Var}\big(\big[\mathbb{E}\{\widehat{f}_{\mathrm{ridge}}(x_0)\} - \mathbb{E}\{\widetilde{f}_{\mathrm{ridge}}(x_0)\} + o_P(\zeta(m))\big] + \big\{\phi_2^{(2)}(\widehat{\theta}; x_0) - \widetilde{\phi}_2^{(2)}(\widehat{\theta}; x_0)\big\}\big) \\
=\,&\mathrm{Var}\big(o_P(\zeta(m)) + \big\{\phi_2^{(2)}(\widehat{\theta}; x_0) - \widetilde{\phi}_2^{(2)}(\widehat{\theta}; x_0)\big\}\big) \\
=\,&\mathrm{Var}\big(\phi_2^{(2)}(\widehat{\theta}; x_0) - \widetilde{\phi}_2^{(2)}(\widehat{\theta}; x_0)\big) + o_P(\zeta(m)) \\
=\,&\mathrm{Var}\big(\phi_2^{(2)}(\widehat{\theta}; x_0)\big) + \mathrm{Var}\big(\widetilde{\phi}_2^{(2)}(\widehat{\theta}; x_0)\big) - 2\mathrm{Cov}\big(\phi_2^{(2)}(\widehat{\theta}; x_0), \widetilde{\phi}_2^{(2)}(\widehat{\theta}; x_0)\big) + o_P(\zeta(m)) \\
\to\,&\sigma^2(x_0) + \widetilde{\sigma}^2(x_0) - 2\rho\widetilde{\sigma}(x_0)\sigma(x_0) + o_P(\zeta(m)).
\end{aligned}
$$

## A.6   Addtional Information for Experiments

### A.6.1   Real Data Descriptions

**Dataset Information**: The datasets BIO, BIKE, DIAMOND, and STOCK have feature dimension d=9, and CT has original dimension d=379 (reduced to 50 features through correlation-based selection).

**Auxiliary Data Description**: The auxiliary datasets are generated through three distinct approaches: (1) Data partitioning by feature: for the DIAMOND dataset, we select samples with the highest clarity as the training and test set (representing rare, high-cost-to-label instances) while using lower-clarity samples as auxiliary data, creating significant distributional heterogeneity; (2) Shared-origin

data with label corruption: for CT and STOCK datasets, the target and auxliary initially share the same distribution, but we systematically corrupt parts of the auxiliary data (10% relabeling from $N(\mu_{\text{CT}}, \sigma^2_{\text{CT}})$ for CT, 40% from $N(\mu_{\text{STOCK}}, \sigma^2_{\text{STOCK}})$ for STOCK) to simulate annotation/transmission errors or data missing, where $\mu_{\text{CT}}, \mu_{\text{STOCK}}$ and $\sigma^2_{\text{CT}}, \sigma^2_{\text{STOCK}}$ represent the expectation and variance of response variable in CT, STOCK dataset; (3) Feature-dependent label perturbation: for BIO and BIKE datasets, we modify auxiliary data $(x, y)$ to be $(x, y_{\text{new}}), y_{\text{new}} = y + \delta(x)$, where $\delta(x)$ models instrumentation or recording errors. For BIO dataset, $\delta(x) = \cos(\sum_{i=1}^{d} x_i)$ and for BIKE dataset $\delta(x) = 2\cos(\sum_{i=1}^{d} x_i)$. Table 3 conclusively demonstrates significant distributional shifts between auxiliary and target data.

### A.6.2 Additional Results

In this section we provide additional and detailed results for our experiments.

Table 2: Comparison of 8 selection methods across synthetic and real-world datasets in terms of RMSE at step 10, where Avg Impro represents improvement over BMDAL averaged across 7 experiments.

|  | S1 | S2 | BIO | BIKE | DIAMOND | CT | STOCK | Avg Impro |
|---|---|---|---|---|---|---|---|---|
| random | 0.890 | 1.352 | 0.412 | 0.295 | 17.854 | 0.308 | 0.346 | |
| lcmd | 0.971 | 1.443 | 0.390 | 0.256 | 16.644 | 0.168 | 0.328 | |
| lcmd (ours) | 0.803 | 1.206 | 0.392 | 0.291 | 17.045 | 0.197 | **0.315**$^*$ | 0.6% |
| maxdist | 0.833 | 1.255 | 0.404 | 0.285 | 16.691 | 0.177 | 0.343 | |
| maxdist (ours) | 0.797 | 1.202 | 0.386 | 0.272 | 16.657 | 0.186 | 0.321 | 2.7% |
| kmeanspp | 0.856 | 1.307 | 0.389 | 0.259 | 16.139 | 0.181 | 0.327 | |
| kmeanspp (ours) | 0.808 | 1.230 | **0.381**$^*$ | **0.246**$^*$ | **15.974**$^*$ | 0.176 | 0.318 | 3.5% |
| fw | 0.915 | 1.382 | 0.407 | 0.295 | 17.582 | 0.272 | 0.341 | |
| fw (ours) | 0.856 | 1.268 | 0.391 | 0.273 | 18.389 | 0.231 | 0.341 | 5.2% |
| bait | 0.812 | 1.267 | 0.404 | 0.307 | 18.218 | 0.375 | 0.350 | |
| bait (ours) | **0.797**$^*$ | 1.227 | 0.398 | 0.271 | 19.114 | 0.299 | 0.335 | 5.4% |
| maxdet | 0.843 | 1.260 | 0.393 | 0.333 | 16.257 | 0.240 | 0.336 | |
| maxdet (ours) | 0.802 | **1.200**$^*$ | 0.383 | 0.258 | 16.587 | **0.161**$^*$ | 0.328 | 9.8% |
| maxdiag | 0.880 | 1.363 | 0.426 | 0.431 | 20.508 | 0.451 | 0.372 | |
| maxdiag (ours) | 0.800 | 1.207 | 0.388 | 0.280 | 16.223 | 0.196 | 0.323 | 22.2% |

Table 3: RMSE comparison at step 15: target-only trained versus target and auxiliary combined data (Aux) trained models.

|  | S1 | S2 | BIO | BIKE | DIAMOND | CT | STOCK |
|---|---|---|---|---|---|---|---|
| maxdiag | 0.819 | 1.205 | 0.384 | 0.233 | 14.029 | 0.150 | 0.306 |
| maxdiag (Aux) | 1.514 | 1.958 | 0.639 | 1.008 | 19.256 | 0.199 | 0.325 |
| maxdet | 0.819 | 1.223 | 0.374 | 0.223 | 13.817 | 0.131 | 0.309 |
| maxdet (Aux) | 1.512 | 1.956 | 0.631 | 1.048 | 19.215 | 0.193 | 0.337 |
| bait | 0.818 | 1.254 | 0.396 | 0.230 | 17.131 | 0.200 | 0.309 |
| bait (Aux) | 1.527 | 1.977 | 0.628 | 1.056 | 19.610 | 0.248 | 0.329 |
| fw | 0.889 | 1.332 | 0.381 | 0.219 | 16.957 | 0.178 | 0.317 |
| fw (Aux) | 1.547 | 1.985 | 0.620 | 1.096 | 19.165 | 0.213 | 0.329 |
| maxdist | 0.829 | 1.223 | 0.384 | 0.230 | 13.675 | 0.141 | 0.304 |
| maxdist (Aux) | 1.528 | 1.967 | 0.636 | 1.003 | 19.161 | 0.200 | 0.337 |
| kmeanspp | 0.831 | 1.244 | 0.378 | 0.214 | 13.708 | 0.140 | 0.301 |
| kmeanspp (Aux) | 1.521 | 1.973 | 0.625 | 1.012 | 19.155 | 0.199 | 0.330 |
| lcmd | 0.837 | 1.235 | 0.383 | 0.240 | 14.421 | 0.148 | 0.301 |
| lcmd (Aux) | 1.529 | 1.964 | 0.635 | 1.004 | 19.191 | 0.198 | 0.338 |
| random | 1.011 | 1.490 | 0.397 | 0.235 | 16.307 | 0.233 | 0.319 |
| random (Aux) | 1.541 | 1.994 | 0.622 | 1.077 | 19.455 | 0.257 | 0.333 |

### A.6.3 Additional Computational Burden Analysis Results

For different methods, in Table 11 we present the computational resource consumption, where AvgUsg denotes the memory usage and AvgTime represents the average runtime per step of the method. The first row reports the consumption of BMDAL, followed by the consumption of our method under

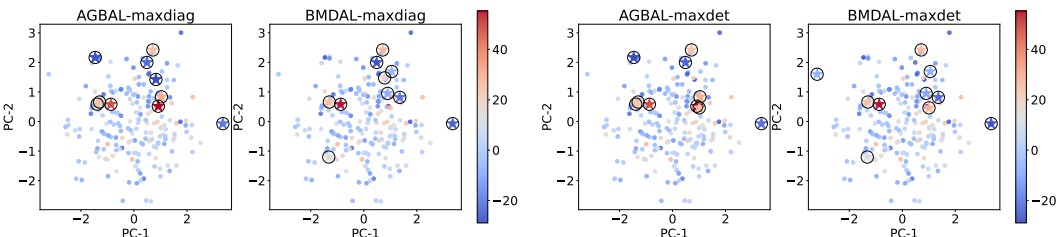

Figure 5: Visualization of the loss of selected points for maxdiag and maxdet methods.

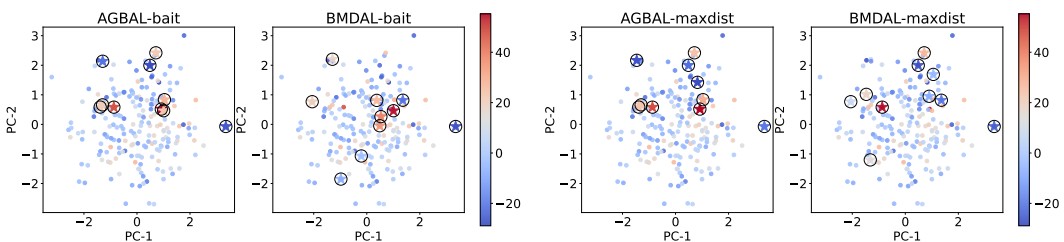

Figure 6: Visualization of the loss of selected points for bait and maxdist methods.

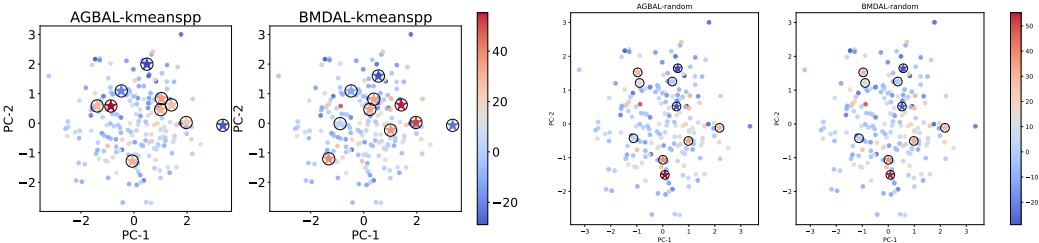

Figure 7: Visualization of the loss of selected points for kmeanspp and random methods.

Table 4: MSE results of S1.

|  | 0 | 1 | 2 | 3 | 4 | 5 | 6 | 7 | 8 | 9 | 10 | 11 | 12 | 13 | 14 | 15 |
|---|---|---|---|---|---|---|---|---|---|---|---|---|---|---|---|---|
| random | 1.077 | 1.058 | 0.993 | 0.971 | 0.955 | 0.942 | 0.930 | 0.921 | 0.911 | 0.900 | 0.890 | 0.882 | 0.875 | 0.868 | 0.862 | 0.857 |
| maxdiag | 1.077 | 0.972 | 0.928 | 0.921 | 0.917 | 0.912 | 0.908 | 0.900 | 0.894 | 0.886 | 0.880 | 0.873 | 0.867 | 0.862 | 0.857 | 0.853 |
| maxdiag (ours) | **1.077** | **0.977** | **0.879** | **0.859** | **0.843** | **0.834** | **0.826** | **0.818** | **0.810** | **0.805** | **0.800** | **0.795** | **0.791** | **0.788** | **0.786** | **0.783** |
| maxdet | 1.077 | 0.982 | 0.924 | 0.906 | 0.893 | 0.881 | 0.874 | 0.866 | 0.858 | 0.850 | 0.843 | 0.837 | 0.831 | 0.826 | 0.821 | 0.818 |
| maxdet (ours) | **1.077** | **0.949** | **0.867** | **0.846** | **0.838** | **0.828** | **0.822** | **0.817** | **0.811** | **0.806** | **0.802** | **0.798** | **0.795** | **0.792** | **0.790** | **0.788** |
| bait | 1.077 | 1.014 | 0.917 | 0.885 | 0.864 | 0.850 | 0.839 | 0.831 | 0.824 | 0.818 | 0.812 | 0.807 | 0.802 | 0.799 | 0.796 | 0.794 |
| bait (ours) | **1.077** | **1.000** | **0.874** | **0.853** | **0.842** | **0.830** | **0.822** | **0.814** | **0.808** | **0.802** | **0.797** | **0.793** | **0.788** | **0.785** | **0.783** | **0.781** |
| fw | 1.077 | 1.062 | 1.016 | 0.995 | 0.987 | 0.978 | 0.967 | 0.953 | 0.939 | 0.926 | 0.915 | 0.906 | 0.898 | 0.891 | 0.884 | 0.879 |
| fw (ours) | **1.077** | **1.046** | **0.963** | **0.942** | **0.930** | **0.915** | **0.899** | **0.887** | **0.876** | **0.865** | **0.856** | **0.847** | **0.840** | **0.834** | **0.829** | **0.825** |
| maxdist | 1.077 | 0.963 | 0.900 | 0.884 | 0.875 | 0.867 | 0.859 | 0.851 | 0.845 | 0.839 | 0.833 | 0.826 | 0.822 | 0.817 | 0.814 | 0.810 |
| maxdist (ours) | **1.077** | **0.966** | **0.881** | **0.861** | **0.846** | **0.833** | **0.824** | **0.816** | **0.808** | **0.802** | **0.797** | **0.793** | **0.789** | **0.786** | **0.783** | **0.781** |
| kmeanspp | 1.077 | 1.020 | 0.954 | 0.926 | 0.915 | 0.906 | 0.896 | 0.888 | 0.876 | 0.865 | 0.856 | 0.848 | 0.841 | 0.835 | 0.831 | 0.827 |
| kmeanspp (ours) | **1.077** | **0.967** | **0.892** | **0.864** | **0.849** | **0.838** | **0.831** | **0.825** | **0.819** | **0.813** | **0.808** | **0.803** | **0.799** | **0.795** | **0.792** | **0.790** |
| lcmd | 1.077 | 1.082 | 1.076 | 1.084 | 1.075 | 1.066 | 1.046 | 1.027 | 1.010 | 0.988 | 0.971 | 0.956 | 0.942 | 0.931 | 0.921 | 0.913 |
| lcmd (ours) | **1.077** | **0.988** | **0.948** | **0.876** | **0.853** | **0.840** | **0.831** | **0.822** | **0.815** | **0.808** | **0.803** | **0.798** | **0.794** | **0.791** | **0.788** | **0.786** |

Table 5: MSE results of S2.

| | 0 | 1 | 2 | 3 | 4 | 5 | 6 | 7 | 8 | 9 | 10 | 11 | 12 | 13 | 14 | 15 |
|---|---|---|---|---|---|---|---|---|---|---|---|---|---|---|---|---|
| random | 1.703 | 1.652 | 1.541 | 1.492 | 1.460 | 1.441 | 1.424 | 1.403 | 1.384 | 1.367 | 1.352 | 1.339 | 1.328 | 1.318 | 1.310 | 1.303 |
| maxdiag | 1.703 | 1.516 | 1.451 | 1.427 | 1.423 | 1.415 | 1.409 | 1.395 | 1.383 | 1.373 | 1.363 | 1.352 | 1.342 | 1.334 | 1.326 | 1.319 |
| maxdiag (ours) | **1.703** | **1.504** | **1.346** | **1.309** | **1.279** | **1.261** | **1.246** | **1.235** | **1.225** | **1.216** | **1.207** | **1.200** | **1.195** | **1.191** | **1.187** | **1.184** |
| maxdet | 1.703 | 1.492 | 1.402 | 1.364 | 1.341 | 1.324 | 1.308 | 1.295 | 1.283 | 1.270 | 1.260 | 1.251 | 1.244 | 1.238 | 1.233 | 1.228 |
| maxdet (ours) | **1.703** | **1.453** | **1.310** | **1.277** | **1.259** | **1.245** | **1.231** | **1.221** | **1.212** | **1.206** | **1.200** | **1.195** | **1.190** | **1.186** | **1.182** | **1.180** |
| bait | 1.703 | 1.632 | 1.459 | 1.396 | 1.360 | 1.336 | 1.319 | 1.303 | 1.290 | 1.278 | 1.267 | 1.258 | 1.250 | 1.244 | 1.238 | 1.234 |
| bait (ours) | **1.703** | **1.568** | **1.382** | **1.326** | **1.303** | **1.287** | **1.270** | **1.256** | **1.245** | **1.235** | **1.227** | **1.220** | **1.214** | **1.209** | **1.205** | **1.201** |
| fw | 1.703 | 1.663 | 1.571 | 1.528 | 1.496 | 1.473 | 1.451 | 1.434 | 1.415 | 1.398 | 1.382 | 1.367 | 1.355 | 1.344 | 1.335 | 1.328 |
| fw (ours) | **1.703** | **1.586** | **1.459** | **1.408** | **1.376** | **1.350** | **1.329** | **1.311** | **1.296** | **1.282** | **1.268** | **1.257** | **1.248** | **1.241** | **1.234** | **1.229** |
| maxdist | 1.703 | 1.491 | 1.390 | 1.348 | 1.322 | 1.306 | 1.292 | 1.281 | 1.271 | 1.263 | 1.255 | 1.248 | 1.243 | 1.238 | 1.233 | 1.230 |
| maxdist (ours) | **1.703** | **1.495** | **1.358** | **1.304** | **1.273** | **1.255** | **1.241** | **1.226** | **1.216** | **1.208** | **1.202** | **1.196** | **1.192** | **1.188** | **1.185** | **1.182** |
| kmeanspp | 1.703 | 1.601 | 1.502 | 1.454 | 1.417 | 1.393 | 1.374 | 1.354 | 1.336 | 1.321 | 1.307 | 1.295 | 1.285 | 1.276 | 1.270 | 1.264 |
| kmeanspp (ours) | **1.703** | **1.532** | **1.383** | **1.336** | **1.308** | **1.291** | **1.276** | **1.263** | **1.249** | **1.239** | **1.230** | **1.222** | **1.216** | **1.210** | **1.205** | **1.202** |
| lcmd | 1.703 | 1.748 | 1.686 | 1.607 | 1.582 | 1.563 | 1.538 | 1.511 | 1.486 | 1.466 | 1.443 | 1.423 | 1.405 | 1.390 | 1.376 | 1.365 |
| lcmd (ours) | **1.703** | **1.534** | **1.422** | **1.326** | **1.291** | **1.270** | **1.254** | **1.239** | **1.224** | **1.215** | **1.206** | **1.198** | **1.193** | **1.188** | **1.183** | **1.180** |

Table 6: MSE results of BIO.

| | 0 | 1 | 2 | 3 | 4 | 5 | 6 | 7 | 8 | 9 | 10 | 11 | 12 | 13 | 14 | 15 |
|---|---|---|---|---|---|---|---|---|---|---|---|---|---|---|---|---|
| random | 0.677 | 0.536 | 0.512 | 0.485 | 0.468 | 0.468 | 0.449 | 0.435 | 0.428 | 0.416 | 0.412 | 0.409 | 0.405 | 0.401 | 0.399 | 0.397 |
| maxdiag | 0.677 | 0.525 | 0.490 | 0.472 | 0.457 | 0.449 | 0.447 | 0.442 | 0.435 | 0.430 | 0.426 | 0.420 | 0.415 | 0.412 | 0.409 | 0.406 |
| maxdiag (ours) | **0.677** | **0.502** | **0.468** | **0.470** | **0.451** | **0.463** | **0.487** | **0.503** | **0.503** | **0.515** | **0.519** | **0.523** | **0.524** | **0.526** | **0.533** | **0.537** |
| maxdet | 0.677 | 0.472 | 0.448 | 0.434 | 0.418 | 0.412 | 0.407 | 0.402 | 0.399 | 0.397 | 0.393 | 0.391 | 0.389 | 0.388 | 0.387 | 0.386 |
| maxdet (ours) | **0.677** | **0.483** | **0.443** | **0.418** | **0.408** | **0.400** | **0.395** | **0.391** | **0.387** | **0.385** | **0.383** | **0.382** | **0.381** | **0.380** | **0.379** | **0.380** |
| bait | 0.677 | 0.558 | 0.503 | 0.467 | 0.448 | 0.439 | 0.429 | 0.424 | 0.412 | 0.409 | 0.404 | 0.401 | 0.398 | 0.397 | 0.394 | 0.392 |
| bait (ours) | **0.677** | **0.513** | **0.478** | **0.460** | **0.439** | **0.429** | **0.422** | **0.413** | **0.407** | **0.402** | **0.398** | **0.396** | **0.395** | **0.393** | **0.390** | **0.388** |
| fw | 0.677 | 0.519 | 0.470 | 0.457 | 0.437 | 0.430 | 0.422 | 0.415 | 0.412 | 0.411 | 0.407 | 0.403 | 0.400 | 0.398 | 0.396 | 0.394 |
| fw (ours) | **0.677** | **0.484** | **0.451** | **0.434** | **0.419** | **0.411** | **0.406** | **0.400** | **0.398** | **0.394** | **0.391** | **0.391** | **0.389** | **0.388** | **0.387** | **0.386** |
| maxdist | 0.677 | 0.486 | 0.460 | 0.448 | 0.433 | 0.423 | 0.419 | 0.414 | 0.409 | 0.406 | 0.404 | 0.401 | 0.398 | 0.396 | 0.395 | 0.394 |
| maxdist (ours) | **0.677** | **0.501** | **0.466** | **0.440** | **0.421** | **0.408** | **0.400** | **0.394** | **0.391** | **0.388** | **0.386** | **0.384** | **0.382** | **0.381** | **0.379** | **0.378** |
| kmeanspp | 0.677 | 0.474 | 0.443 | 0.429 | 0.415 | 0.406 | 0.401 | 0.396 | 0.393 | 0.390 | 0.389 | 0.387 | 0.385 | 0.384 | 0.382 | 0.382 |
| kmeanspp (ours) | **0.677** | **0.467** | **0.431** | **0.417** | **0.407** | **0.399** | **0.394** | **0.390** | **0.385** | **0.384** | **0.381** | **0.379** | **0.378** | **0.377** | **0.376** | **0.375** |
| lcmd | 0.677 | 0.472 | 0.452 | 0.432 | 0.422 | 0.415 | 0.405 | 0.400 | 0.397 | 0.393 | 0.390 | 0.388 | 0.386 | 0.385 | 0.384 | 0.383 |
| lcmd (ours) | **0.677** | **0.472** | **0.461** | **0.450** | **0.432** | **0.421** | **0.412** | **0.402** | **0.396** | **0.394** | **0.392** | **0.390** | **0.387** | **0.385** | **0.384** | **0.383** |

Table 7: MSE results of BIKE.

| | 0 | 1 | 2 | 3 | 4 | 5 | 6 | 7 | 8 | 9 | 10 | 11 | 12 | 13 | 14 | 15 |
|---|---|---|---|---|---|---|---|---|---|---|---|---|---|---|---|---|
| random | 1.413 | 0.923 | 0.764 | 0.613 | 0.537 | 0.462 | 0.408 | 0.368 | 0.340 | 0.315 | 0.295 | 0.278 | 0.263 | 0.252 | 0.243 | 0.235 |
| maxdiag | 1.413 | 1.051 | 0.929 | 0.798 | 0.705 | 0.622 | 0.574 | 0.524 | 0.483 | 0.454 | 0.431 | 0.409 | 0.389 | 0.367 | 0.349 | 0.332 |
| maxdiag (ours) | **1.413** | **0.830** | **0.620** | **0.505** | **0.426** | **0.378** | **0.349** | **0.324** | **0.307** | **0.292** | **0.280** | **0.269** | **0.259** | **0.249** | **0.242** | **0.236** |
| maxdet | 1.413 | 0.890 | 0.751 | 0.631 | 0.561 | 0.504 | 0.452 | 0.413 | 0.381 | 0.356 | 0.333 | 0.315 | 0.300 | 0.286 | 0.276 | 0.268 |
| maxdet (ours) | **1.413** | **0.698** | **0.476** | **0.397** | **0.351** | **0.327** | **0.312** | **0.291** | **0.280** | **0.268** | **0.258** | **0.249** | **0.242** | **0.237** | **0.231** | **0.226** |
| bait | 1.413 | 0.971 | 0.789 | 0.661 | 0.574 | 0.491 | 0.431 | 0.390 | 0.355 | 0.328 | 0.307 | 0.292 | 0.278 | 0.267 | 0.255 | 0.246 |
| bait (ours) | **1.413** | **0.775** | **0.589** | **0.480** | **0.416** | **0.374** | **0.346** | **0.321** | **0.301** | **0.285** | **0.271** | **0.261** | **0.251** | **0.242** | **0.235** | **0.230** |
| fw | 1.413 | 0.947 | 0.745 | 0.614 | 0.514 | 0.443 | 0.399 | 0.365 | 0.335 | 0.312 | 0.295 | 0.278 | 0.263 | 0.253 | 0.245 | 0.237 |
| fw (ours) | **1.413** | **0.809** | **0.577** | **0.491** | **0.429** | **0.390** | **0.357** | **0.330** | **0.308** | **0.289** | **0.273** | **0.261** | **0.251** | **0.243** | **0.236** | **0.229** |
| maxdist | 1.413 | 0.915 | 0.716 | 0.579 | 0.486 | 0.423 | 0.377 | 0.346 | 0.321 | 0.300 | 0.285 | 0.271 | 0.259 | 0.249 | 0.240 | 0.232 |
| maxdist (ours) | **1.413** | **0.829** | **0.597** | **0.484** | **0.412** | **0.368** | **0.340** | **0.316** | **0.300** | **0.285** | **0.272** | **0.260** | **0.251** | **0.242** | **0.234** | **0.227** |
| kmeanspp | 1.413 | 0.852 | 0.654 | 0.511 | 0.431 | 0.380 | 0.342 | 0.313 | 0.288 | 0.273 | 0.259 | 0.249 | 0.239 | 0.231 | 0.224 | 0.219 |
| kmeanspp (ours) | **1.413** | **0.762** | **0.522** | **0.416** | **0.362** | **0.327** | **0.300** | **0.283** | **0.268** | **0.257** | **0.246** | **0.238** | **0.230** | **0.224** | **0.218** | **0.213** |
| lcmd | 1.413 | 0.819 | 0.620 | 0.502 | 0.420 | 0.369 | 0.331 | 0.302 | 0.281 | 0.266 | 0.256 | 0.245 | 0.237 | 0.229 | 0.223 | 0.218 |
| lcmd (ours) | **1.413** | **0.819** | **0.620** | **0.559** | **0.489** | **0.450** | **0.411** | **0.369** | **0.341** | **0.317** | **0.291** | **0.276** | **0.261** | **0.252** | **0.242** | **0.234** |

Table 8: MSE results of DIAMOND.

| | 0 | 1 | 2 | 3 | 4 | 5 | 6 | 7 | 8 | 9 | 10 | 11 | 12 | 13 | 14 | 15 |
|---|---|---|---|---|---|---|---|---|---|---|---|---|---|---|---|---|
| random | 33.666 | 29.685 | 26.771 | 25.483 | 23.523 | 21.760 | 20.474 | 19.429 | 18.923 | 18.359 | 17.854 | 17.398 | 17.095 | 16.809 | 16.582 | 16.307 |
| maxdiag | 33.666 | 34.604 | 33.749 | 30.630 | 27.929 | 26.469 | 24.681 | 23.864 | 22.387 | 21.282 | 20.508 | 19.791 | 19.487 | 19.085 | 18.491 | 18.232 |
| maxdiag (ours) | **33.666** | **35.427** | **27.964** | **25.435** | **23.301** | **22.054** | **20.700** | **19.560** | **18.331** | **16.854** | **16.223** | **15.840** | **15.467** | **15.038** | **14.841** | **14.628** |
| maxdet | 33.666 | 30.538 | 25.993 | 23.145 | 21.704 | 20.044 | 19.326 | 18.241 | 17.625 | 16.713 | 16.257 | 15.887 | 15.462 | 15.276 | 15.040 | 14.882 |
| maxdet (ours) | **33.666** | **31.333** | **26.014** | **24.047** | **21.857** | **20.019** | **18.761** | **18.211** | **17.210** | **17.039** | **16.587** | **16.273** | **15.725** | **15.288** | **15.081** | **14.826** |
| bait | 33.666 | 31.433 | 28.557 | 26.973 | 25.002 | 23.365 | 22.640 | 20.566 | 19.652 | 18.897 | 18.218 | 18.127 | 17.578 | 17.345 | 17.183 | 16.977 |
| bait (ours) | **33.666** | **31.764** | **28.619** | **26.016** | **24.163** | **22.536** | **21.492** | **20.835** | **20.311** | **19.377** | **19.114** | **18.677** | **18.249** | **17.802** | **17.557** | **17.320** |
| fw | 33.666 | 31.308 | 29.604 | 26.492 | 23.034 | 21.262 | 20.625 | 19.736 | 18.715 | 17.968 | 17.582 | 16.979 | 16.695 | 16.216 | 15.871 | 15.608 |
| fw (ours) | **33.666** | **31.662** | **28.651** | **26.862** | **24.885** | **23.475** | **21.564** | **20.582** | **19.677** | **19.077** | **18.389** | **17.633** | **17.213** | **16.733** | **16.413** | **15.914** |
| maxdist | 33.666 | 33.311 | 27.960 | 24.517 | 22.498 | 21.050 | 19.973 | 19.212 | 18.238 | 17.297 | 16.691 | 16.460 | 16.095 | 15.988 | 15.775 | 15.498 |
| maxdist (ours) | **33.666** | **36.023** | **29.659** | **26.134** | **23.642** | **22.181** | **20.896** | **19.661** | **18.201** | **17.259** | **16.657** | **16.290** | **15.980** | **15.466** | **15.346** | **14.930** |
| kmeanspp | 33.666 | 30.066 | 26.346 | 23.903 | 21.836 | 20.200 | 19.375 | 17.934 | 17.055 | 16.410 | 16.139 | 15.869 | 15.617 | 15.309 | 15.029 | 14.880 |
| kmeanspp (ours) | **33.666** | **31.448** | **26.154** | **22.890** | **21.410** | **20.365** | **19.123** | **18.308** | **17.335** | **16.468** | **15.974** | **15.656** | **15.188** | **15.075** | **14.715** | **14.523** |
| lcmd | 33.666 | 30.379 | 26.531 | 23.069 | 21.023 | 19.593 | 18.864 | 18.003 | 17.329 | 17.037 | 16.644 | 16.159 | 15.914 | 15.502 | 15.322 | 15.008 |
| lcmd (ours) | **33.666** | **30.379** | **26.501** | **26.659** | **22.573** | **21.850** | **21.215** | **19.384** | **18.087** | **17.673** | **17.045** | **16.671** | **16.032** | **16.048** | **15.667** | **15.366** |

Table 9: MSE results of CT.

| | 0 | 1 | 2 | 3 | 4 | 5 | 6 | 7 | 8 | 9 | 10 | 11 | 12 | 13 | 14 | 15 |
|---|---|---|---|---|---|---|---|---|---|---|---|---|---|---|---|---|
| random | 0.848 | 0.595 | 0.503 | 0.445 | 0.419 | 0.399 | 0.372 | 0.358 | 0.351 | 0.337 | 0.308 | 0.292 | 0.276 | 0.258 | 0.245 | 0.233 |
| maxdiag | 0.848 | 0.641 | 0.608 | 0.597 | 0.593 | 0.594 | 0.598 | 0.564 | 0.538 | 0.503 | 0.451 | 0.431 | 0.407 | 0.387 | 0.371 | 0.351 |
| maxdiag (ours) | **0.848** | **0.595** | **0.457** | **0.411** | **0.379** | **0.331** | **0.295** | **0.266** | **0.236** | **0.215** | **0.196** | **0.183** | **0.173** | **0.166** | **0.158** | **0.153** |
| maxdet | 0.848 | 0.527 | 0.440 | 0.391 | 0.364 | 0.342 | 0.318 | 0.299 | 0.278 | 0.259 | 0.240 | 0.224 | 0.211 | 0.202 | 0.193 | 0.184 |
| maxdet (ours) | **0.848** | **0.517** | **0.394** | **0.333** | **0.286** | **0.260** | **0.230** | **0.203** | **0.186** | **0.171** | **0.161** | **0.153** | **0.146** | **0.140** | **0.135** | **0.131** |
| bait | 0.848 | 0.620 | 0.554 | 0.514 | 0.470 | 0.462 | 0.438 | 0.431 | 0.411 | 0.393 | 0.375 | 0.356 | 0.327 | 0.313 | 0.298 | 0.289 |
| bait (ours) | **0.848** | **0.620** | **0.488** | **0.454** | **0.425** | **0.395** | **0.378** | **0.359** | **0.354** | **0.322** | **0.299** | **0.277** | **0.255** | **0.234** | **0.225** | **0.211** |
| fw | 0.848 | 0.554 | 0.463 | 0.417 | 0.388 | 0.367 | 0.355 | 0.330 | 0.299 | 0.286 | 0.272 | 0.259 | 0.241 | 0.227 | 0.216 | 0.207 |
| fw (ours) | **0.848** | **0.498** | **0.430** | **0.391** | **0.346** | **0.323** | **0.303** | **0.286** | **0.266** | **0.249** | **0.231** | **0.216** | **0.206** | **0.196** | **0.187** | **0.179** |
| maxdist | 0.848 | 0.538 | 0.415 | 0.356 | 0.309 | 0.277 | 0.250 | 0.225 | 0.205 | 0.190 | 0.177 | 0.168 | 0.157 | 0.150 | 0.143 | 0.138 |
| maxdist (ours) | **0.848** | **0.610** | **0.467** | **0.407** | **0.363** | **0.328** | **0.291** | **0.264** | **0.232** | **0.204** | **0.186** | **0.171** | **0.160** | **0.151** | **0.146** | **0.142** |
| kmeanspp | 0.848 | 0.507 | 0.415 | 0.356 | 0.311 | 0.280 | 0.252 | 0.232 | 0.214 | 0.194 | 0.181 | 0.168 | 0.159 | 0.152 | 0.146 | 0.139 |
| kmeanspp (ours) | **0.848** | **0.510** | **0.399** | **0.335** | **0.292** | **0.269** | **0.247** | **0.222** | **0.206** | **0.189** | **0.176** | **0.166** | **0.157** | **0.149** | **0.143** | **0.138** |
| lcmd | 0.848 | 0.485 | 0.371 | 0.331 | 0.293 | 0.266 | 0.244 | 0.217 | 0.198 | 0.179 | 0.168 | 0.158 | 0.148 | 0.140 | 0.135 | 0.132 |
| lcmd (ours) | **0.848** | **0.485** | **0.433** | **0.392** | **0.354** | **0.321** | **0.283** | **0.257** | **0.242** | **0.216** | **0.197** | **0.185** | **0.174** | **0.165** | **0.157** | **0.151** |

Table 10: MSE results of STOCK.

| | 0 | 1 | 2 | 3 | 4 | 5 | 6 | 7 | 8 | 9 | 10 | 11 | 12 | 13 | 14 | 15 |
|---|---|---|---|---|---|---|---|---|---|---|---|---|---|---|---|---|
| random | 0.588 | 0.498 | 0.468 | 0.451 | 0.430 | 0.409 | 0.389 | 0.379 | 0.367 | 0.355 | 0.346 | 0.340 | 0.334 | 0.328 | 0.324 | 0.319 |
| maxdiag | 0.588 | 0.541 | 0.496 | 0.465 | 0.448 | 0.429 | 0.416 | 0.400 | 0.392 | 0.383 | 0.372 | 0.365 | 0.358 | 0.352 | 0.346 | 0.342 |
| maxdiag (ours) | **0.588** | **0.476** | **0.434** | **0.395** | **0.374** | **0.360** | **0.349** | **0.339** | **0.333** | **0.328** | **0.323** | **0.319** | **0.315** | **0.312** | **0.309** | **0.307** |
| maxdet | 0.588 | 0.483 | 0.456 | 0.429 | 0.405 | 0.387 | 0.372 | 0.361 | 0.350 | 0.343 | 0.336 | 0.330 | 0.324 | 0.321 | 0.316 | 0.313 |
| maxdet (ours) | **0.588** | **0.449** | **0.407** | **0.388** | **0.367** | **0.357** | **0.350** | **0.341** | **0.337** | **0.331** | **0.328** | **0.325** | **0.322** | **0.319** | **0.316** | **0.313** |
| bait | 0.588 | 0.495 | 0.468 | 0.447 | 0.424 | 0.404 | 0.388 | 0.377 | 0.369 | 0.360 | 0.350 | 0.343 | 0.336 | 0.331 | 0.326 | 0.321 |
| bait (ours) | **0.588** | **0.479** | **0.431** | **0.406** | **0.387** | **0.378** | **0.365** | **0.355** | **0.349** | **0.342** | **0.335** | **0.329** | **0.324** | **0.321** | **0.317** | **0.312** |
| fw | 0.588 | 0.513 | 0.477 | 0.442 | 0.421 | 0.404 | 0.385 | 0.368 | 0.360 | 0.351 | 0.341 | 0.336 | 0.330 | 0.324 | 0.319 | 0.316 |
| fw (ours) | **0.588** | **0.480** | **0.445** | **0.416** | **0.392** | **0.382** | **0.369** | **0.360** | **0.352** | **0.345** | **0.341** | **0.335** | **0.330** | **0.327** | **0.323** | **0.319** |
| maxdist | 0.588 | 0.511 | 0.481 | 0.445 | 0.416 | 0.402 | 0.386 | 0.372 | 0.362 | 0.350 | 0.343 | 0.337 | 0.330 | 0.325 | 0.321 | 0.317 |
| maxdist (ours) | **0.588** | **0.477** | **0.436** | **0.400** | **0.375** | **0.359** | **0.345** | **0.341** | **0.333** | **0.327** | **0.321** | **0.318** | **0.315** | **0.312** | **0.309** | **0.306** |
| kmeanspp | 0.588 | 0.493 | 0.457 | 0.430 | 0.398 | 0.382 | 0.365 | 0.353 | 0.342 | 0.334 | 0.327 | 0.320 | 0.315 | 0.310 | 0.306 | 0.303 |
| kmeanspp (ours) | **0.588** | **0.452** | **0.412** | **0.389** | **0.371** | **0.357** | **0.346** | **0.338** | **0.329** | **0.324** | **0.318** | **0.314** | **0.311** | **0.308** | **0.305** | **0.302** |
| lcmd | 0.588 | 0.498 | 0.452 | 0.421 | 0.396 | 0.374 | 0.358 | 0.347 | 0.339 | 0.332 | 0.328 | 0.321 | 0.316 | 0.312 | 0.308 | 0.305 |
| lcmd (ours) | **0.588** | **0.498** | **0.464** | **0.419** | **0.392** | **0.365** | **0.345** | **0.332** | **0.325** | **0.319** | **0.315** | **0.311** | **0.308** | **0.306** | **0.303** | **0.301** |

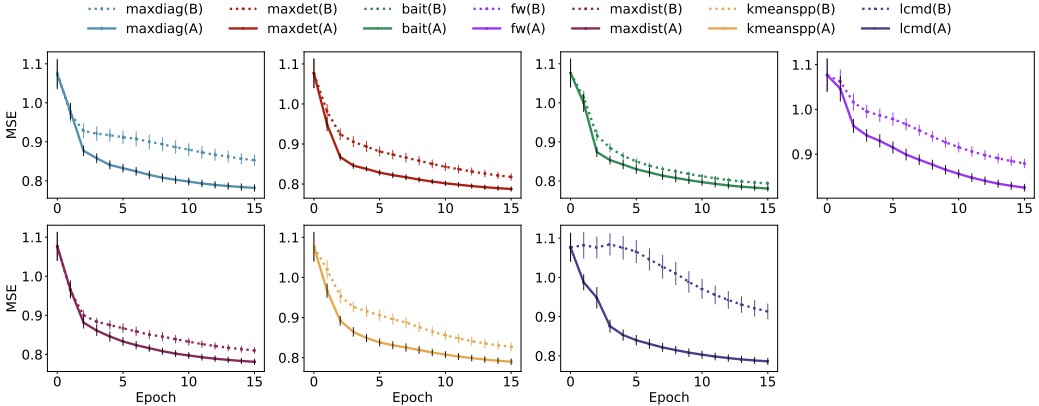

Figure 8: S1 MSEs plot.

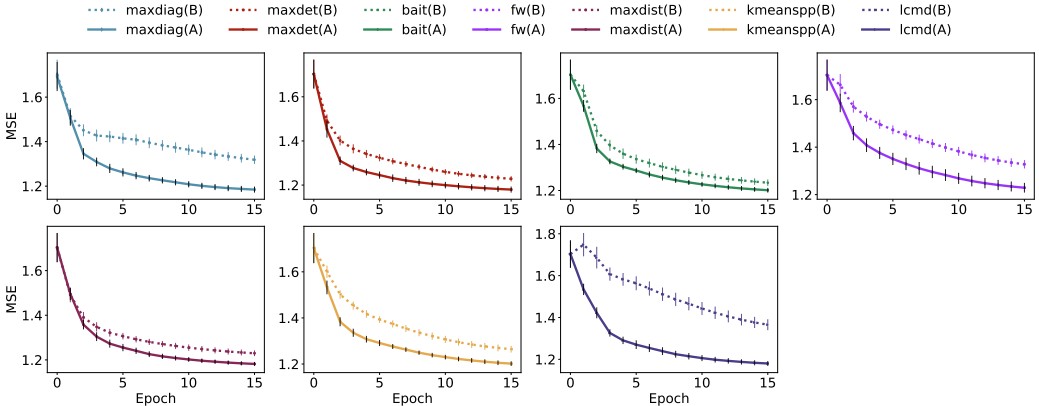

Figure 9: S2 MSEs plot.

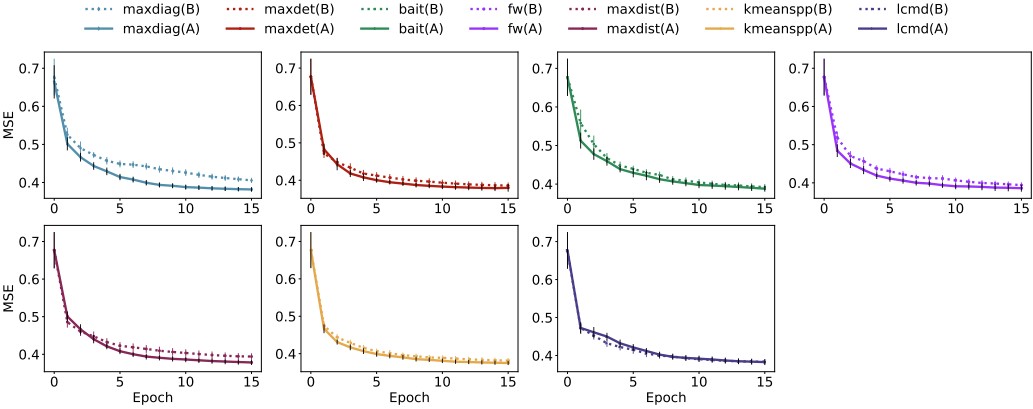

Figure 10: BIO MSEs plot.

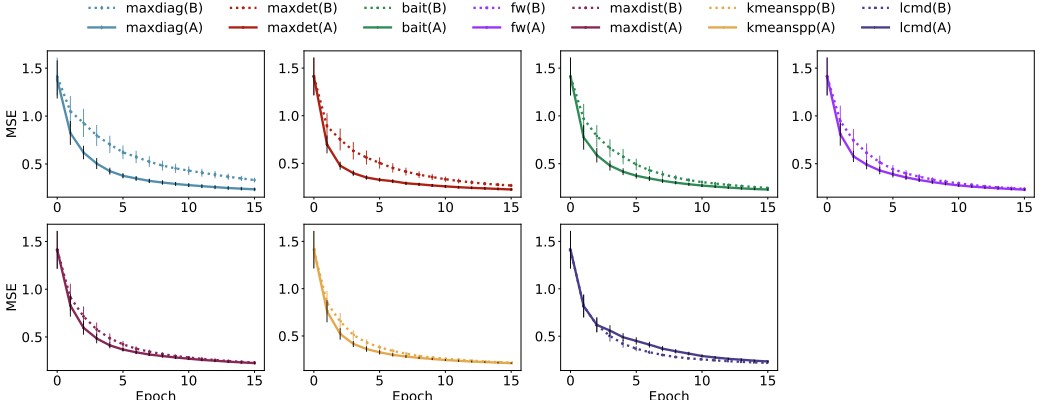

Figure 11: BIKE MSEs plot.

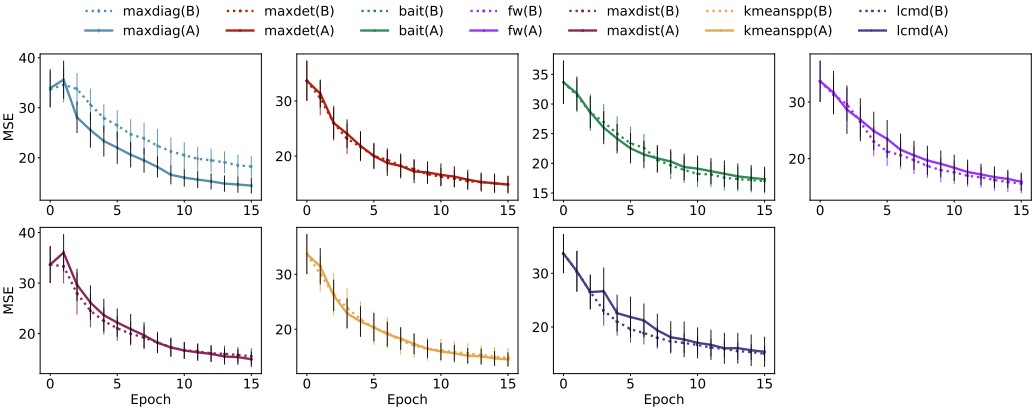

Figure 12: DIAMOND MSEs plot.

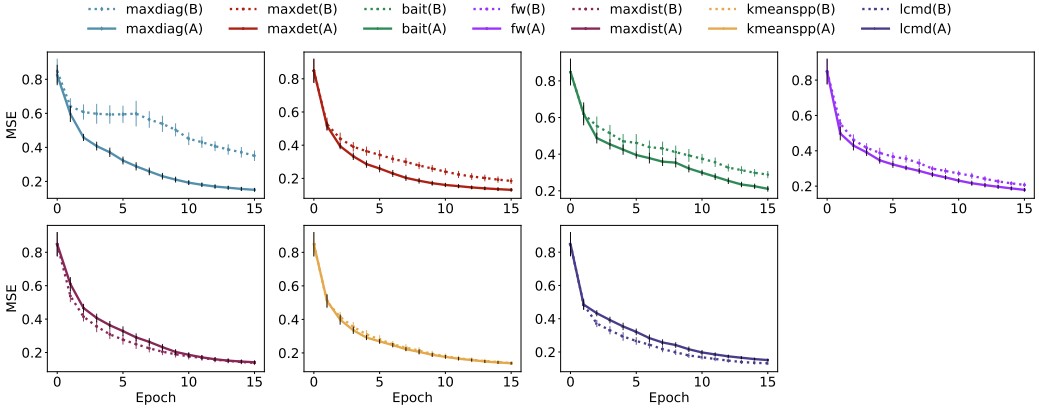

Figure 13: CT MSEs plot.

varying sizes of auxiliary data samples, along with the corresponding AUC improvement achieved at each sample size.

Table 11: Computational Burden Analysis Results.

|  |  | lcmd | maxdiag | maxdet | fw | maxdist | bait | kmeanspp |
|---|---|---|---|---|---|---|---|---|
| AvgUsg (M) | BMDAL | 571.4 | 502.2 | 553.8 | 544.0 | 576.0 | 596.6 | 562.2 |
|  | ours (100) | 575.2 | 501.4 | 561.4 | 570.6 | 570.0 | 599.0 | 511.0 |
|  | ours (500) | 593.4 | 522.0 | 578.6 | 567.0 | 587.0 | 609.0 | 520.2 |
|  | ours (1000) | 595.0 | 529.0 | 584.8 | 578.0 | 585.8 | 610.8 | 539.2 |
|  | ours (10000) | 766.0 | 713.4 | 750.0 | 752.4 | 770.6 | 769.2 | 765.4 |
| AvgTime (s) | BMDAL | 1.39 | 0.06 | 0.32 | 0.23 | 1.19 | 1.10 | 1.28 |
|  | ours (100) | 2.35 | 0.91 | 1.23 | 1.12 | 2.10 | 2.02 | 2.41 |
|  | ours (500) | 3.08 | 1.62 | 1.97 | 1.83 | 2.80 | 2.72 | 2.81 |
|  | ours (1000) | 3.57 | 2.27 | 2.62 | 2.46 | 3.39 | 3.32 | 3.17 |
|  | ours (10000) | 15.04 | 14.49 | 15.23 | 14.74 | 15.95 | 15.44 | 15.89 |
| AvgImpro (%) | ours (100) | −11.47 | 14.91 | **5.83** | 2.28 | −9.17 | 5.86 | −0.66 |
|  | ours (500) | −6.17 | 20.31 | **10.69** | 4.66 | −0.40 | 8.95 | 2.60 |
|  | ours (1000) | −5.65 | 17.99 | **10.47** | 5.03 | 1.13 | 7.71 | 3.86 |
|  | ours (10000) | −6.08 | 22.56 | **12.02** | 5.20 | 1.91 | 8.76 | 4.47 |

## A.7 Additional Experiments

### A.7.1 Experiments on Auxiliary Data Quality

Since the core of our method lies in improving loss estimation through auxiliary data guidance, an intuitive expectation is that better alignment between auxiliary and target distributions should yield superior estimation performance. This naturally raises the question: how severely will our method degrade as auxiliary data quality deteriorates? To investigate this, we adopt the same experimental settings as defined in S1 and S2.

At $\zeta = 64$, where the distribution shift is most severe, AGBAL achieves its highest AUC for the MSE curve while still outperforming BMDAL (which uses no auxiliary data). As shown in Table 12, AGBAL maintains superior performance across most selection methods, except for a slight degradation under maxdist. This demonstrates our method's robustness: when distribution discrepancy becomes extreme, the density ratio estimation effectively nullifies the auxiliary data's influence (weights approach zero), preventing negative learning while maintaining comparable performance to not using auxiliary data at all.

Table 12: Worst case AUC comparison between AGBAL and BMDAL.

|  |  | maxdiag | maxdet | bait | fw | maxdist | kmeanspp | lcmd |
|---|---|---|---|---|---|---|---|---|
| S1 | BMDAL | 0.956 | 0.952 | 0.914 | 1.038 | 0.933 | 0.975 | 1.131 |
|  | AGBAL (ours) | 0.942 | 0.918 | 0.904 | 1.038 | 0.947 | 0.940 | 0.975 |
|  | Improvement | 1.5% | 3.6% | 1.1% | 0.0% | −1.5% | 3.6% | 13.8% |
| S2 | BMDAL | 1.501 | 1.430 | 1.417 | 1.583 | 1.406 | 1.479 | 1.647 |
|  | AGBAL (ours) | 1.437 | 1.398 | 1.390 | 1.543 | 1.436 | 1.426 | 1.468 |
|  | Improvement | 4.3% | 2.2% | 1.9% | 2.5% | −2.1% | 3.6% | 10.9% |

### A.7.2 Experiments on Auxiliary Data Quantity

On the other hand, the volume of auxiliary data warrants investigation. While real-world scenarios typically provide abundant auxiliary data (e.g., corrupted data or related-task data), limited-quantity cases do exist - such as small historical archives with long-term records. To examine the performance of our method with scarce auxiliary data, we conduct experiments with varying volumes $(50, 100, 300, 500, 1000$ samples) under the S1 and S2 configurations, fixing $\zeta = 8$. For each selection method, we evaluate the performance by computing the AUC of MSE curves.

Figure 14-15 presents the AUC values of MSE curves for AGBAL (solid lines) versus BMDAL (dashed lines, without auxiliary data) across varying auxiliary data sizes ($N_{\text{aux}}$). The results demon-

strate: (1) AGBAL's performance improves monotonically with increasing $N_{\mathrm{aux}}$; (2) At minimal data volume ($N_{\mathrm{aux}} = 50$), AGBAL achieves comparable performance to BMDAL while maintaining superiority in most cases; (3) The only exception occurs under maxdist selection, where AGBAL marginally underperforms BMDAL. These findings confirm that even with modest amounts of auxiliary data, AGBAL can be still useful.

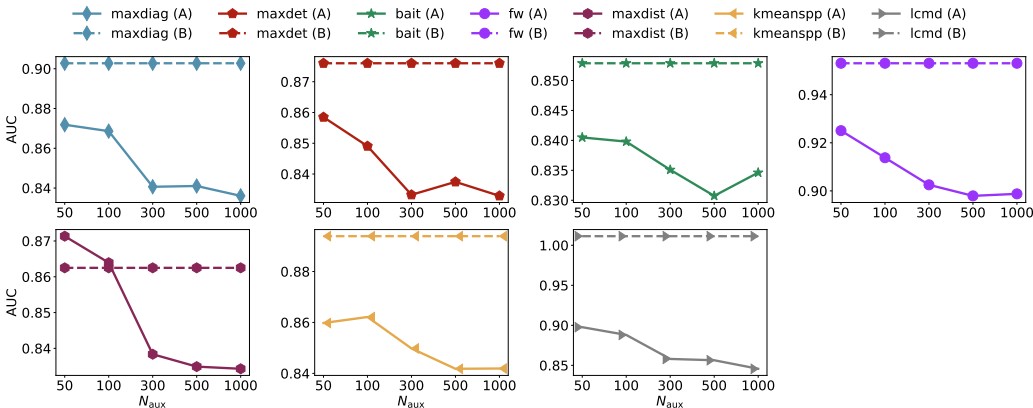

Figure 14: AUC plots of varying $N_{\mathrm{aux}}$ for S1.

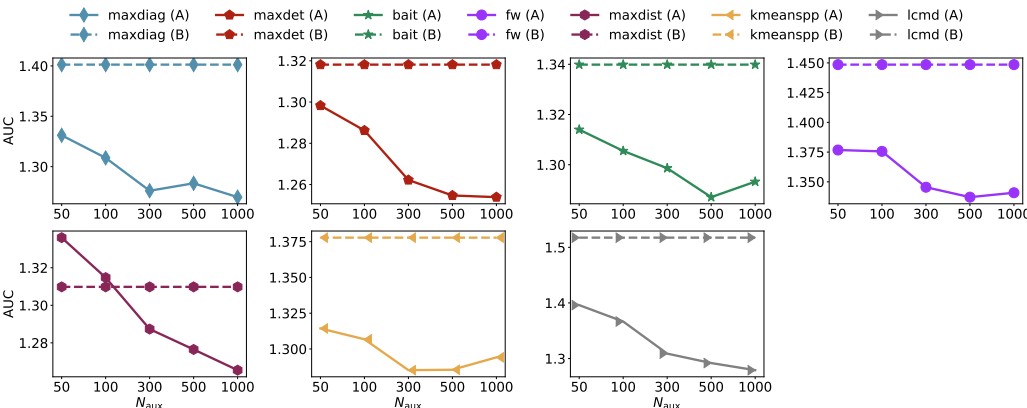

Figure 15: AUC plots of varying $N_{\mathrm{aux}}$ for S2.

We evaluate our method on five real-world datasets with varying auxiliary data sizes $N_{\mathrm{aux}}$. Figure 16-20 illustrates the AUC trends against auxiliary data volume. Due to the inherent distributional complexity of real-world data, the AUC-n curves exhibit non-smooth variations. However, aggregating results across all five datasets and selection methods reveals: (1) a consistent decreasing trend in AUC as $N_{\mathrm{aux}}$ increases; (2) at minimal auxiliary data ($N_{\mathrm{aux}} = 100$), while AGBAL's advantage over BMDAL (measured by optimal selection method AUC) becomes marginal, it remains competitive - demonstrating the framework's robustness.

## A.8 Limitations

The performance of AGBAL is fundamentally constrained by the quality and relevance of the auxiliary data. While the method demonstrates robustness to moderate distribution shifts between auxiliary and target distributions, its effectiveness diminishes when the auxiliary data becomes too noisy or exhibits systematic biases.

The theoretical analysis relies on NTK assumptions that may not hold in practical settings. While the infinite-width network approximation provides valuable insights, its applicability to modern deep architectures with finite width and complex layer interactions remains uncertain. This paper does

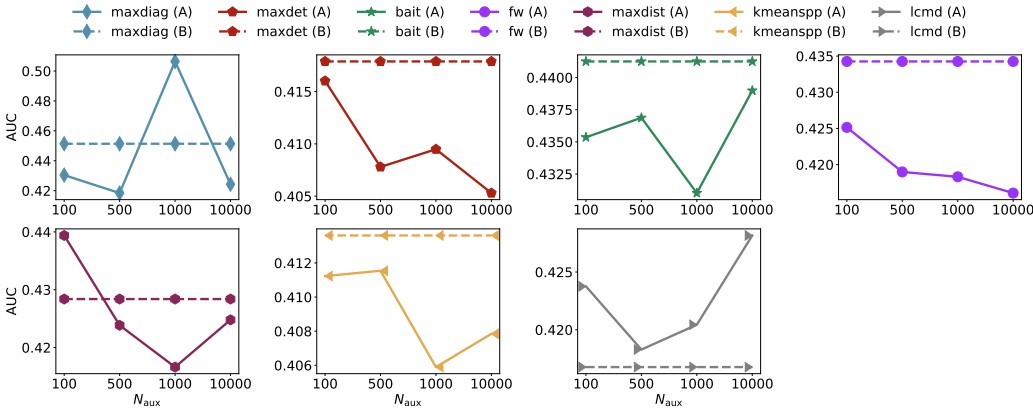

Figure 16: AUC plots of varying $N_{\mathrm{aux}}$ for BIO.

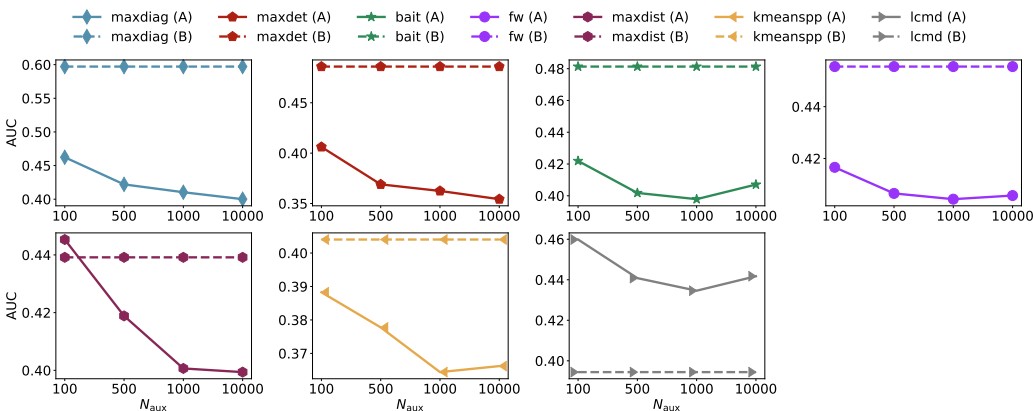

Figure 17: AUC plots of varying $N_{\mathrm{aux}}$ for BIKE.

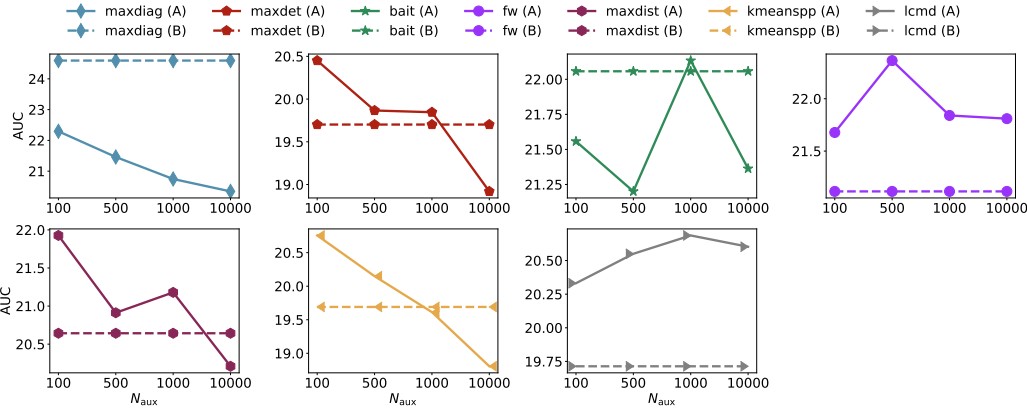

Figure 18: AUC plots of varying $N_{\mathrm{aux}}$ for DIAMOND.

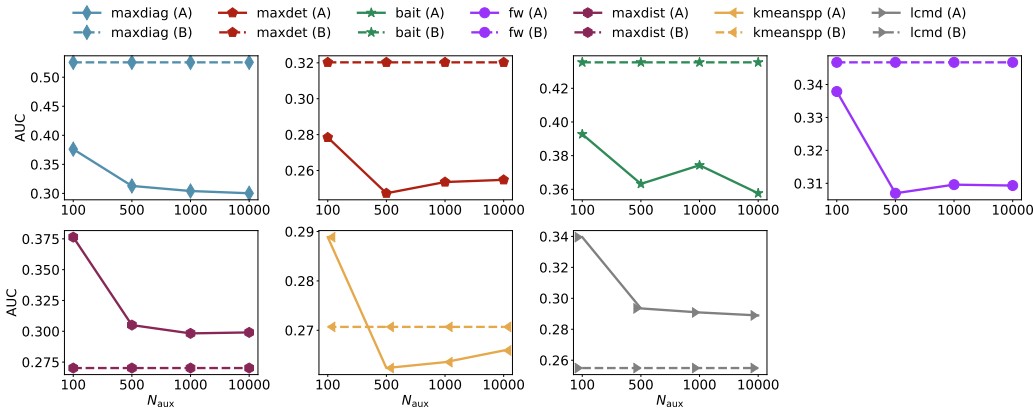

Figure 19: AUC plots of varying $N_{\mathrm{aux}}$ for CT.

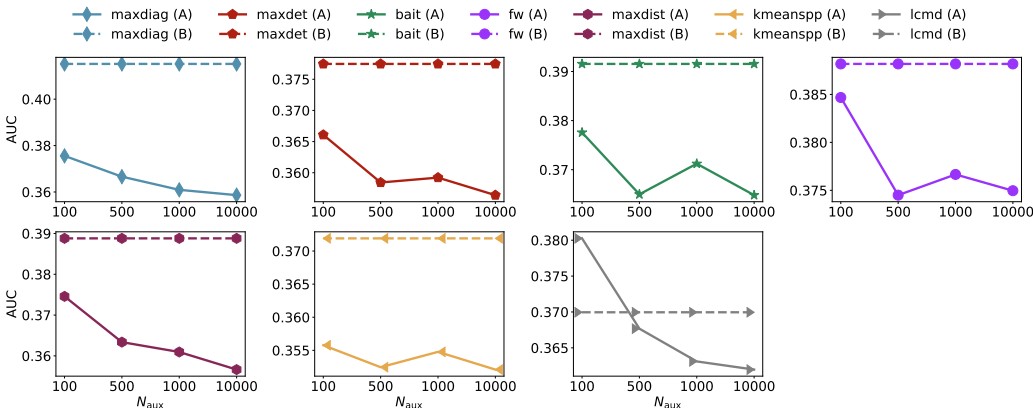

Figure 20: AUC plots of varying $N_{\mathrm{aux}}$ for STOCK.

not investigate how deviations from these ideal conditions might affect the method's performance in real-world applications.

The experimental validation, while comprehensive, focuses primarily on regression tasks with relatively low-dimensional input. The performance of AGBAL on high-dimensional structured data, such as images or time series, remains unexplored. Additionally, all experiments assume the availability of auxiliary data with similar feature spaces, leaving open the question of how the method would perform when auxiliary data come from substantially different modalities.

### A.9 Broader Impacts

The proposed AGBAL framework has several potential positive social impacts. By reducing annotation costs through more efficient active learning, our method could make machine learning more accessible in resource-constrained domains such as healthcare in developing regions or small-scale industrial applications. The ability to leverage imperfect auxiliary data aligns well with real-world scenarios where perfect datasets are rare, potentially enabling more applications in safety-critical domains such as medical diagnosis or autonomous driving.

While effective, our method raises privacy concerns when auxiliary data contains sensitive information, and may require fairness considerations when adapted to classification tasks. Though efficiency gains could lower barriers for misuse, this risk is mitigated by the method's domain-agnostic nature. We encourage future work on privacy-preserving and fairness-aware extensions.

