# OpenReview forum: "Enhancing Deep Batch Active Learning for Regression with Imperfect Data Guided Selection"
_NeurIPS.cc/2025/Conference — NeurIPS 2025 poster_

### Official Review · Reviewer_AvdT · 2025-06-06

**Clarity:** 4
**Significance:** 3
**Originality:** 3
**Rating:** 5
**Confidence:** 3

**Summary:**

This paper proposes an active learning approach that leverages auxiliary data (data that is distributionally shifted but relevant) to help with predicting sample uncertainty. The approach uses this data to estimate density ratio between auxiliary and target data distributions, compute model's loss on auxiliary data, and then apply density ratio weighting to obtain approximation of true loss. The paper then verifies their approach through theoretical analysis based on Neural Tangent Kernel theory. Results show strong performance of their approch.

**Questions:**

1. How does your method compare against prior works that also use auxiliary data in active learning?
2. How does your method compare against baselines in terms of compute resources and time?
3. Are there any sort of analysis or arguments that can be extended to understand how the proposed approach may translate to real-world applications?

**Ethical Concerns:**

["NO or VERY MINOR ethics concerns only"]

**Final Justification:**

I keep my same score of Accept, as I believe this paper brings novel contributions to active learning field.

**Limitations:**

yes

**Quality:**

4

**Strengths And Weaknesses:**

Strengths:
* Intuitive yet innovative active learning approach to using auxiliary data
* Well-supported analysis of their proposed approach
* Writing and motivation is clear throughout the paper
* Empirical results show consistent improvement with their proposed approach and appendix has very thorough experimentation ablations

Weaknesses:
* Lack of mentions/comparison against prior work that also make use of auxiliary data for active learning (eg "Active Transfer Learning under Model Shift" [2014])
* As mentioned in paper's limitations, unsure how NTK analysis translates to real-world applications as it assumes infinite-width network approximation
* Lack of clarity on experiments compute resource. While the paper runs many experiments, it is unclear how costly their proposed approach is in terms of compute resources and time.

---

> ### Author Rebuttal · Authors · 2025-07-29
>
> We sincerely appreciate your positive feedback highlighting our innovative yet intuitive approach to leveraging auxiliary data in active learning, along with your recognition of our rigorous theoretical analysis, clear presentation, and thorough experimental validation including detailed ablation studies. Below we address all comments to further strengthen these contributions.
>
> **W1\&Q1:** We sincerely appreciate the reviewer's insightful comment regarding the comparison with transfer active learning methods. Our initial submission lacked explicit discussion of this important aspect, and we thank the reviewer for bringing this to our attention.
> - Our work differs from transfer learning approaches like [1] in the definition of auxiliary data:
>     - transfer learning and domain adaptation assume covariate shift with different $X$ distribution but consistent distribution of $P(Y\mid X)$ between source and target domains [1];
>     - our setting considers imperfect auxiliary data where both $P(X)$ and $P(Y\mid X)$ may differ.
> - This distinction is crucial because standard transfer methods could suffer from negative transfer under such joint distribution shift. To validate this empirically, we compared directly combining target and auxiliary data versus our approach, with results of our approach demonstrating superior performance (reported in Table 3 of our original submission).
> - We agree that exploring connections to transfer active learning methods would be valuable, and we will include this discussion in the revised version, along with potential extensions combining our method with transfer techniques when the $P(Y\mid X)$ consistency holds. This clarification will better position our contribution relative to prior work.
>
> **W2/Q3:** We sincerely appreciate the reviewer's thoughtful question regarding the practical implications of our NTK-based analysis.
> - While our current theoretical analysis employs the infinite-width neural network approximation for tractability, we note that prior work [2] has established rigorous connections between finite-width networks and their neural tangent kernel behavior, suggesting our theoretical insights could be extended to practical settings. However, such extension would require substantial additional technical details.
> - The infinite-width analysis serves to provide theoretical justification for our approach, while empirical results demonstrate the method's performance with finite-width networks across multiple real-world datasets. We agree that further investigation of finite-width cases would be valuable and will consider this direction in future work.
>
> **W3/Q2:** We greatly appreciate the reviewer's important question regarding the computational efficiency of our method.
> - Our approach introduces computational overhead through density ratio estimation and auxiliary loss computation, but we mitigate this through careful design choices:
>     - employing efficient machine learning methods for density ratio and loss estimation;
>     - allowing flexible control of auxiliary dataset sizes.
> - The selection period maintains the same computational complexity as BMDAL.
> - Empirical results demonstrate that our method achieves meaningful performance improvements even with modest auxiliary data sizes (100, 500, 1000 samples), where the additional computational overhead remains negligible. With only 100 auxiliary data, the maxdet selection strategy, which performs best among the compared approaches, still outperforms BMDAL.
> - All experiments were conducted on a standard server (dual Intel Xeon Gold 6330 CPUs, 125GB RAM), with detailed time/memory comparisons provided in the table below. The performance gains consistently justify the modest computational cost, particularly in active learning settings where annotation costs typically dominate computation costs.
>
> We will include the computational analysis in the revised version.
>
> |||lcmd|maxdiag|maxdet|fw|maxdist|bait|kmeanspp|
> |-|-|-|-|-|-|-|-|-|
> |**AvgUsg(M)**|BMDAL|571.40|502.20|553.80|544.00|576.00|596.60|562.20|
> ||ours(100)|575.20|501.40|561.40|570.60|570.00|599.00|511.00|
> ||ours(500)|593.40|522.00|578.60|567.00|587.00|609.00|520.20|
> ||ours(1000)|595.00|529.00|584.80|578.00|585.80|610.80|539.20|
> ||ours(10000)|766.00|713.40|750.00|752.40|770.60|769.20|765.40|
> |**AvgTime(s)**|BMDAL|1.39|0.06|0.32|0.23|1.19|1.10|1.28|
> ||ours(100)|2.35|0.91|1.23|1.12|2.10|2.02|2.41|
> ||ours(500)|3.08|1.62|1.97|1.83|2.80|2.72|2.81|
> ||ours(1000)|3.57|2.27|2.62|2.46|3.39|3.32|3.17|
> ||ours(10000)|15.04|14.49|15.23|14.74|15.95|15.44|15.89|
> |**AvgImpro(%)**|ours(100)|-11.47|14.91|**5.83**|2.28|-9.17|5.86|-0.66|
> ||ours(500)|-6.17|20.31|**10.69**|4.66|-0.40|8.95|2.60|
> ||ours(1000)|-5.65|17.99|**10.47**|5.03|1.13|7.71|3.86|
> ||ours(10000)|-6.08|22.56|**12.02**|5.20|1.91|8.76|4.47|
>
> *We sincerely appreciate your thoughtful and constructive feedback. Your observations have helped pinpoint several significant areas for future study. We are more than happy to answer any further questions during the rebuttal period.*
>
> **Reference**
>
> [1] Wang, Xuezhi, Tzu-Kuo Huang, and Jeff Schneider. "Active transfer learning under model shift." International Conference on Machine Learning. PMLR, 2014.
>
> [2] Novak, Roman, Jascha Sohl-Dickstein, and Samuel S. Schoenholz. "Fast finite width neural tangent kernel." International Conference on Machine Learning. PMLR, 2022.

---

> > ### Comment · Reviewer_AvdT · 2025-08-05
> >
> > Thank you for the detailed rebuttal and addressing my previous points. The relation between this work and the previous work on active transfer learning is clear. Would recommend putting that in the paper.
> >
> > For future work,  further investigation of finite-width cases and improving computational efficiency would be good directions.

---

> > > ### Author Response · Authors · 2025-08-05
> > >
> > > Thank you for your thoughtful review and for recognizing our efforts in addressing your previous points. We sincerely appreciate your time and constructive feedback, which has helped strengthen our work. We will clarify the connection to active transfer learning in the revised version and highlight investigation of finite-width cases and computational efficiency as key future work. Your insights have provided valuable guidance for both the current paper and our ongoing research directions!

---

### Official Review · Reviewer_6X3n · 2025-07-02

**Clarity:** 2
**Significance:** 2
**Originality:** 4
**Rating:** 4
**Confidence:** 3

**Summary:**

The paper addresses a long standing problem that in order to know to the gradient for gradients which are required for active learning one usually needs to know the labels if not relaying on strong model assumptions. The paper attempts to solve this egg and chicken problem using similarly, but different, auxiliary data, for which the labels are known. Trough that, the problem is transferred from knowing $p(x,y)$ to knowing $r(x,y)\coloneqq \frac{p(x,y)}{q(x,y)}$.

**Questions:**

1. Why we need to assume $\hat{r}$ is independent of $X_i',Y_i' \sim Q$ (line 153) and why is it reasonable  to assume so.
2. Line 90: (a) Is $\mathcal A$ a function that takes in its second argument an **infinite** training set from $(\mathcal{X}\times\mathcal{Y})^\mathbb{N}$; (b) why does $\mathcal A$ needs the initial parameter in its first argument, how is it been used in the rest of the paper?
3. Why do we need to expect that the estimation.
4. Line 162: why do we expect $\rho$ to be small?
5. What is the  limit of $\mathbb{E}[\hat{\phi}_2(\hat{\theta},x_0)]$ ?
6. Table 1: why is lower AUC better (usually larger is better)?

**Ethical Concerns:**

["NO or VERY MINOR ethics concerns only"]

**Final Justification:**

Comment:
I decided to raise my score.

My main concerns were:

(1) Although the paper presents itself as proposing an active learning method for deep networks, it is essentially focused on active learning for kernel methods.

(2) Some assumptions were unclear.

(3) General clarity.

Although the response only partially addressed my concerns, I eventually raised the score due to the originality and the interesting analysis.

**Limitations:**

Yes, in the checklist.

**Paper Formatting Concerns:**

no concerns

**Quality:**

3

**Strengths And Weaknesses:**

**Strengths**
1. The problem is important and the mathematics looks sound (I did not check proofs).
2. The core idea is original interesting.
3. Experimental results are encouraging.

**Weaknesses**
1. It is not clear for me what is the meaning of the theoretical result, what it guarantees and in which practical conditions.
2. The theoretical result is only applicable for in the infinite width regime.
3. The paper only deals with the square loss.

(Points 2 and 3 are pointed out as  limitations by the authors in the checklist)

---

> ### Author Rebuttal · Authors · 2025-07-29
>
> We are grateful for your valuable assessment of our work's significance and mathematical soundness, along with your appreciation of the core idea's originality and the encouraging experimental results. Below we provide detailed responses to all comments to further strengthen these aspects.
>
> **W1:** Thank you for your thoughtful question regarding the theoretical guarantees of our method.
> - Our theoretical result builds upon NTK theory to establish that in infinitely wide networks, the estimation error $\phi_2^{(2)}$ at each point $x$ stems from the variance $\sigma^2(x)$ of NTK kernel regression on the target sample set. The key insight is that when we have two low-correlation estimates $\phi_2^{(2)}$ and $\widetilde{\phi}_2^{(2)}$ derived from this variance, the variance of their difference becomes approximately proportional to $\sigma^2(x)$. This allows us to use the difference between two estimates as a proxy for estimation error.
> - The theoretical guarantee holds under practical conditions where (i) the auxiliary data provides reasonably accurate estimates (ensured by prior knowledge about information relevance), and (ii) the auxiliary data is either independent or exhibits low correlation with the target data. These conditions are satisfied by our definition of auxiliary data.
>
> **W2:** Thank you for your insightful comment regarding the applicability of our theoretical results. While our current analysis is presented in the infinite-width regime, we note that theoretical treatments of finite-width neural tangent kernels already exist in prior work [1], which contains extensive technical derivations. Our analysis could be extended to the finite-width case following [1], but this would introduce substantial additional complexity. Since our primary contribution focuses on the active learning framework rather than NTK theory itself, we have chosen to present the cleaner infinite-width case. We will consider expanding this theoretical aspect in future work.
>
> **W3:** We sincerely appreciate the reviewer's observation regarding the loss function used in our analysis.
> - While our theoretical development employs the $L_2$ loss, we emphasize that our core methodology is fundamentally applicable to any differentiable loss function.
> - The $L_2$ loss provides superior mathematical convenience due to its smooth, everywhere-differentiable nature and its analytical tractability.
> - In contrast, non-smooth losses like $L_1$ (whose gradient is discontinuous) introduce significant analytical challenges that would obscure our primary contributions. However, the proposed active learning framework itself remains valid for general loss functions in practice.
>
> **Q1:** We are grateful for the reviewer's careful attention to this technical detail.
> - The equality in lines 154-155 requires $\hat{r}$ to be independent of $X_{i}^{\prime},Y_{i}^{\prime}\sim Q$ to ensure the objective of argmin can be expressed as a sum of independent terms.
> - The assumption is practically reasonable because we assume the availability of abundant auxiliary data. In practice, we can split the auxiliary data into two parts: one subset for estimating the density ratio $\hat{r}$ and another independent subset for loss estimation. This data-splitting strategy satisfies the independence requirement.
>
> **Q2:** Thanks for raising this detail.
> - The training set $L_t$ contains a finite number of samples $\mathbb{N}$, but algorithm $\mathcal{A}$ can incorporate arbitrarily many samples. The current expression was chosen for simplicity. Also, to maintain notational consistency, we require an initialization parameter for the model.
> - The initial parameters can be either 0 or randomly generated from a given distribution. This initialization has two purposes: (1) providing a starting point for the NTK parameters, and (2) following standard machine learning practice where parameter initialization is required for training procedures.
>
> **Q3:** We are not certain which specific expectation the reviewer is referring to. However, if the question concerns the expectation in Equation (1), we note that this operation is formally defined in Line 82 of our paper. The expectation appears because we aim to optimize the model parameters with respect to the true risk $R$ over distribution $P$, whose gradient direction requires this expectation. In practice, since we cannot compute this expectation directly, we resort to empirical estimation. This approach aligns with standard practice in statistical learning theory, where population objectives are approximated through finite samples.
>
> **Q4:** The assumption that $\rho$ should be small stems from our core methodological design where we use the difference between two estimates as a proxy for estimation error (as explained in W1). For this approach to be valid, the two estimates must exhibit sufficiently low correlation, hence we require $\rho$ to be small. When $\rho=1$, the estimates become identical, making their difference vanish . This degenerate case provides no meaningful information.
>
> **Q5:** $\hat{\phi}_2(\hat{\theta},x_0)$ is the estimated loss gradient using auxiliary data. As long as the estimator $\hat{\theta}$ converges to the optimal parameter and the density ratio estimator converges to the true density ratio, the limit of $\mathbb{E}[\hat{\phi}_2(\hat{\theta},x_0)]$  converges to $0$.
>
> **Q6:** In Table 1, Area Under the Curve (AUC) metric is employed to evaluate the overall convergence efficiency of different active learning methods. As specified in line 201 of our paper, the AUC metric is calculated by $\text{AUC}=\sum_{i=1}^T(\xi_{i-1}+\xi_i)/2$, where $\xi_0,\ldots,\xi_T$ represents the sequence of average MSE values obtained by each method. In this context, a lower AUC value indicates superior performance, corresponding to smaller prediction errors and more effective sample selection. Conversely, when evaluating based on accuracy curves (where higher values are desirable), the interpretation of AUC values would be inverted.
>
> *We sincerely appreciate your thoughtful and constructive feedback. Your questions have opened up a broader space for discussion. We are more than happy to answer any further questions during the rebuttal period.*
>
> **Reference**
>
> [1] Novak, Roman, Jascha Sohl-Dickstein, and Samuel S. Schoenholz. "Fast finite width neural tangent kernel." International Conference on Machine Learning. PMLR, 2022.

---

### Official Review · Reviewer_XtfN · 2025-07-03

**Clarity:** 3
**Significance:** 3
**Originality:** 3
**Rating:** 5
**Confidence:** 3

**Summary:**

This paper focuses on deep batch active learning for regression tasks where obtaining labels is typically costly, and the objective is to identify a small set of informative unlabeled instances to label, thereby improving the predictive performance of deep learning models. The active selection process typically relies on measures such as model sensitivity and predictive uncertainty. While model sensitivity can be estimated using only unlabeled data, obtaining reliable uncertainty estimates can be markedly challenged by the scarcity of available labeled data. To address this, the paper proposes leveraging auxiliary data, typically drawn from a different distribution than the target task, to guide the active selection process. Although such auxiliary data cannot be directly used to train deep learning models for the main task of interest, the paper shows that they can be used to efficiently estimate the predictive uncertainty to guide the AL process. The proposed method approximates the model's loss by first estimating the density ratio between the auxiliary and main/target data distributions, then computing the model’s loss on the auxiliary data, and finally applying a density ratio weighting scheme. Numerical tests on both synthetic and real-world datasets demonstrate that incorporating auxiliary data in this manner leads to improved performance over the gradient kernel-based BMDAL method, which does not use auxiliary data. The proposed approach is further supported by theoretical analysis via the notion of neural tangent kernels.

**Questions:**

Please refer to weaknesses W1,W2 above. In addition, some questions are listed below:

Q1. What are the properties of the auxiliary dataset relative to the target dataset of the main task, so that the proposed method can be effective?

Q2. What is the role of the size of the auxiliary dataset in the performance of the proposed approach?

**Ethical Concerns:**

["NO or VERY MINOR ethics concerns only"]

**Final Justification:**

The authors have adequately addressed my comments and concerns. Accordingly, I will raise my rating and recommend acceptance of the paper.

**Limitations:**

Yes

**Quality:**

3

**Strengths And Weaknesses:**

STRENGTHS

(S1) The idea of judiciously leveraging auxiliary data to obtain reliable estimates of the model predictive uncertainty that can readily guide the active selection process seems interesting, intuitive, and effective.

(S2) The background, motivation and key concepts of the proposed method are well presented in general.

(S3) The proposed approach is supported by a well-motivated theoretical analysis.

(S4) The literature review is comprehensive.


WEAKNESSES

(W1) The paper does not provide an analysis or discussion of the computational complexity of the proposed method, and it does not report the running time of the proposed method compared to the baselines that do not process auxiliary data in the numerical evaluation. Since leveraging and processing auxiliary data using the density ratio weighting approach introduces additional computational overhead, in my opinion it is important to discuss this additional complexity and show the running time demonstrating the trade-off between prediction performance and running time.

(W2) The proposed method is not compared against effective model predictive uncertainty estimation approaches that can be used for AL; see e.g. [1], [2]. Such an additional test/comparison both in terms of prediction performance and running time, in my opinion is important to further demonstrate the benefits of the proposed approach.

[1] Lakshminarayanan, B., Pritzel, A., & Blundell, C. (2017). Simple and scalable predictive uncertainty estimation using deep ensembles. Advances in neural information processing systems, 30.

[2] Gal, Y., & Ghahramani, Z. (2016, June). Dropout as a bayesian approximation: Representing model uncertainty in deep learning. In International conference on machine learning (pp. 1050-1059). PMLR.

---

> ### Author Rebuttal · Authors · 2025-07-29
>
> Thank you for your positive assessment of our intuitive approach to leveraging auxiliary data for uncertainty estimation, the clear presentation of motivations and key concepts, the well-founded theoretical analysis, and the comprehensive literature review. Below we address all comments and provide additional results.
>
> **W1:** We sincerely appreciate your valuable feedback regarding the computational complexity analysis of our method.
> - While our approach requires processing auxiliary data through density ratio weighting, we balance computational efficiency through two design choices:
>     - using lightweight machine learning methods (e.g., random forests);
>     - controlling the size of auxiliary datasets.
> - The core selection mechanism preserves the same complexity as gradient-only methods. The additional overhead comes from the density ratio estimation and auxiliary loss computation.
> - Our experiments shown in table below demonstrate that both the required resource and the performance improvement scale with the auxiliary data size. We conducted experiments across 5 real-world datasets with varying auxiliary dataset sizes (100, 500, 1000, 10,000 samples) on a server with dual Intel Xeon Gold 6330 CPUs (112 threads total) and 125GB RAM.
> - Even with only 100 auxiliary data, the maxdet selection strategy, which performs best among the compared approaches, still achieves meaningful performance improvements. For auxiliary datasets below 1,000 samples, the additional time and memory overhead remain negligible.
> - This allows us to flexibly balance performance and computational cost by adjusting the auxiliary data size according to their specific requirements. The consistent improvements over gradient-only baselines justify the additional computation, particularly since active learning typically operates in label-efficient regimes where query costs dominate computational costs.
>
> We'll add these discussions in the revised version.
>
> |||lcmd|maxdiag|maxdet|fw|maxdist|bait|kmeanspp|
> |-|-|-|-|-|-|-|-|-|
> |**AvgUsg(M)**|BMDAL|571.40|502.20|553.80|544.00|576.00|596.60|562.20|
> ||ours(100)|575.20|501.40|561.40|570.60|570.00|599.00|511.00|
> ||ours(500)|593.40|522.00|578.60|567.00|587.00|609.00|520.20|
> ||ours(1000)|595.00|529.00|584.80|578.00|585.80|610.80|539.20|
> ||ours(10000)|766.00|713.40|750.00|752.40|770.60|769.20|765.40|
> |**AvgTime(s)**|BMDAL|1.39|0.06|0.32|0.23|1.19|1.10|1.28|
> ||ours(100)|2.35|0.91|1.23|1.12|2.10|2.02|2.41|
> ||ours(500)|3.08|1.62|1.97|1.83|2.80|2.72|2.81|
> ||ours(1000)|3.57|2.27|2.62|2.46|3.39|3.32|3.17|
> ||ours(10000)|15.04|14.49|15.23|14.74|15.95|15.44|15.89|
> |**AvgImpro(%)**|ours(100)|-11.47|14.91|**5.83**|2.28|-9.17|5.86|-0.66|
> ||ours(500)|-6.17|20.31|**10.69**|4.66|-0.40|8.95|2.60|
> ||ours(1000)|-5.65|17.99|**10.47**|5.03|1.13|7.71|3.86|
> ||ours(10000)|-6.08|22.56|**12.02**|5.20|1.91|8.76|4.47|
>
> **W2:** We sincerely appreciate the reviewer's suggestion to compare with uncertainty-based active learning methods.
> - As shown in the table below, both deep ensemble and dropout methods underperform random sampling in our experiments, while AGBAL demonstrates consistent improvements (in Table 1 of our origin manuscript).
> - This observation aligns with findings in [1], which show that pure uncertainty batch sampling without diversity considerations can perform worse than random selection. The limitations of batch uncertainty-based approaches stem from several factors:
>     1. when selecting batches based solely on uncertainty, the chosen samples may cluster in localized high-uncertainty regions while neglecting other informative areas of the input space [1];
>     2. the fundamental assumption that high uncertainty necessarily correlates with prediction errors does not always hold in practice, particularly when model uncertainty estimates are miscalibrated [2];
>     3. the quality of uncertainty estimates is highly sensitive to architectural choices and hyperparameters (e.g., dropout rates, ensemble size) [2,3], requiring careful tuning that may not generalize across different datasets.
> - In contrast, our gradient-based approach directly optimizes the risk minimization objective, making it more robust to dataset characteristics.
>
> |Method|BIO|BIKE|DIAMOND|CT|STOCK|AvgTime(s)|AvgUsg(M)|
> |-|-|-|-|-|-|-|-|
> |random|0.451|0.459|21.009|0.380|0.391|0.00|451.20|
> |dropout|0.471|0.637|23.252|0.492|0.436|0.02|470.40|
> |ensemble|0.507|0.794|23.244|0.582|0.483|7.71|472.20|
>
> **Q1:** We sincerely appreciate this insightful question regarding the key properties of effective auxiliary datasets for our method.
> - Currently, the selection of suitable auxiliary data relies on prior knowledge, and our experimental setup in Appendix A.6.1 provides concrete examples of how we partition target and auxiliary datasets.
> - Through empirical observations, we find that higher-quality auxiliary datasets exhibit smaller variance in the estimated density ratios $\hat{r}(x',y')$. This can be understood through the lens of distributional shift: when target and auxiliary distributions are identical (0\% shift), $\hat{r}(x',y')=1$ with zero variance. As the distributional discrepancy increases, the density ratios deviate more substantially from 1, leading to greater variance. However, this is only an empirical observation, and developing more rigorous quantitative measures for assessing auxiliary data utility is an important direction for future work.
>
> **Q2:** We sincerely thank the reviewer for highlighting this important question.
> - The size of the auxiliary dataset plays a crucial role in balancing performance and computing efficiency in our approach. The results of the experiment in the W1 table demonstrate that a larger auxiliary data set leads to both better performance and more resource consumption.
> - Notably, even minimal auxiliary dataset (100 samples) improves performance under our maxdet selection strategy, which performs best among the compared approaches. The additional time/memory incurred by auxiliary datasets below 1,000 samples is negligible.
>
> *We sincerely appreciate your thoughtful and constructive feedback. The issues you raised have brought to light important questions. We are more than happy to answer any further questions during the rebuttal period.*
>
> **Reference**
>
> [1] Settles, Burr. "Active learning literature survey." (2009).
>
> [2] Tsymbalov, Evgenii, Maxim Panov, and Alexander Shapeev. "Dropout-based active learning for regression." International conference on analysis of images, social networks and texts. Cham: Springer International Publishing, 2018.
>
> [3] Beluch, William H., et al. "The power of ensembles for active learning in image classification." Proceedings of the IEEE conference on computer vision and pattern recognition. 2018.

---

> > ### Comment · Reviewer_XtfN · 2025-08-07
> > **Response to Rebuttal by Authors**
> >
> > I have carefully reviewed the authors’ response, which adequately addresses my comments and concerns. Accordingly, I will raise my rating and recommend acceptance of the paper.

---

> > > ### Author Response · Authors · 2025-08-07
> > >
> > > We are truly grateful for your positive evaluation and for supporting the acceptance of our paper. Your insightful feedback has been very valuable in enhancing our work!

---

### Official Review · Reviewer_Xdew · 2025-07-03

**Clarity:** 2
**Significance:** 3
**Originality:** 4
**Rating:** 4
**Confidence:** 4

**Summary:**

This paper proposes AGBAL (Auxiliary data Guided Batch Active Learning), a new framework for regression-based active learning that incorporates imperfect auxiliary data to better estimate predictive uncertainty. Unlike prior methods which rely heavily on model gradients alone, AGBAL estimates the expected loss gradient by weighting auxiliary data losses using a density ratio between auxiliary and target distributions. The estimated gradient is then used to define an auxiliary-data-guided gradient kernel, improving sample selection strategies. The method is theoretically grounded with derivations based on NTK (Neural Tangent Kernel) theory and evaluated on synthetic and real-world regression datasets, showing consistent performance gains over conventional approaches.

**Questions:**

1. How robust is AGBAL to mislabeled or low-quality auxiliary data? Can performance degrade catastrophically if auxiliary data is noisy, even if the density ratio is correctly estimated?
2. How is AGBAL fundamentally different from importance sampling or domain adaptation weighting schemes? Could this approach be viewed as an instance of transfer-active learning with a known source domain?
3. Given that Lₜ is actively selected and thus biased, how reliable is it as a reference for estimating the density ratio or surrogate gradients? Would using a small, randomly labeled validation set instead improve stability?
4. Could the auxiliary data be adversarially exploited? Since the method uses them for uncertainty estimation, poorly aligned auxiliary data could potentially misguide the acquisition function.
5. Can this framework be extended to classification tasks, or is it strictly limited to regression? The kernel-based formulation and surrogate gradient approximation might generalize.

**Ethical Concerns:**

["NO or VERY MINOR ethics concerns only"]

**Limitations:**

- Biased Labeled Set (Lₜ): The framework assumes Lₜ to be a reliable proxy for the target distribution, but active learning selection is inherently biased.
- Data Quality Sensitivity: The performance is highly sensitive to the quality of the auxiliary dataset and the correctness of the density ratio estimation.
- Distribution Matching Assumption: The method assumes that the auxiliary distribution is absolutely continuous with respect to the target distribution, which may not hold in real-world cases (e.g., outlier auxiliary sources).

**Quality:**

2

**Strengths And Weaknesses:**

Strengths
- Novelty: The idea of leveraging auxiliary (distribution-shifted) data via density ratio-weighted gradient approximation is conceptually fresh and well-motivated for regression tasks, where predictive uncertainty is hard to estimate without labels.
- Theoretical Rigor: The authors provide solid mathematical grounding, including an NTK-based analysis showing how the auxiliary-guided gradient approximates the true expected loss gradient.
- Empirical Effectiveness: Extensive experiments on multiple datasets and across various selection strategies demonstrate that AGBAL consistently improves MSE and convergence over baseline methods.

Weaknesses
- Dependency on Auxiliary Data Quality: The method assumes the availability of auxiliary data that is not only relevant to the target task but also sufficiently high quality. As shown in the authors’ own ablation study (Fig. 2), performance deteriorates when the auxiliary data distribution is heavily shifted. In practice, ensuring auxiliary data quality may be non-trivial.
- Resemblance to Importance Sampling: The core mechanism of re-weighting auxiliary losses by a density ratio is reminiscent of importance sampling in off-policy learning or domain adaptation. While the authors mention this connection, a clearer differentiation—conceptually or in results—is needed.
- Bias in Target Labeled Set (Lₜ): The estimation of the density ratio relies on the labeled target dataset Lₜ, which is obtained through active learning. However, such data is typically biased, as AL tends to over-sample uncertain or diverse examples. This raises concerns about the validity of the density ratio estimation and the generalizability of the reweighted losses.
- Auxiliary Uncertainty Estimation is Still Model-Dependent: Although the method avoids relying solely on model gradients, the uncertainty estimation through auxiliary loss still depends on a trained model, and errors or biases in that model can propagate into the selection process.
- Borderline Scalability: The full method requires training and maintaining auxiliary loss estimators, density ratio estimators, and selection kernels, which could add computational burden compared to simpler gradient-based AL strategies.
- Readability: Certain sections of the paper-particulary Section 2.3- are difficult to follow. The exposition is mathematically dense and lacks intuition-building commentary. The derivation involving NTK-based decomposition of loss gradient estimation is correct but could benefit from clearer structure, step-wise annotations, or illustrative figures.
- Scalability and Complexity: The full pipeline involves model training, auxiliary data loss computation, density ratio estimation, and guided kernel construction. The practical cost of running AGBAL is not clearly discussed, making it less attractive for low-resource or fast-iteration settings.

---

> ### Author Rebuttal · Authors · 2025-07-29
>
> We sincerely thank the reviewer for their positive assessment recognizing the novelty of our density ratio-weighted gradient method for regression uncertainty, the theoretical rigor of our NTK analysis, and the empirical effectiveness across diverse datasets. Below we address all comments and provide additional results.
>
> **W1\&Q1\&Q4:** Thank you for your thoughtful comment regarding the dependency on auxiliary data quality. We appreciate your attention to this important aspect of our method.
> - As shown in Figure 2, the performance degrades only linearly even when the parameter $\zeta$ controlling the distribution shift increases exponentially. This suggests that our approach is not overly sensitive to moderate mismatches in auxiliary data quality.
> - We acknowledge that when auxiliary data is poorly aligned with the target distribution (e.g., having completely disjoint support), the estimated loss may become uncorrelated with the true loss. In such extreme cases, if the estimated loss is small where the true loss is large, this could indeed misguide the acquisition function.
> - Currently, we determine whether the auxiliary data is suitable based on prior domain knowledge. We fully agree that developing quantitative measures to evaluate auxiliary data quality would be valuable to prevent such adversarial scenarios. This is an important direction we plan to explore in future work including both auxiliary data quality measurement and high-quality auxiliary data selection.
>
> **W2\&Q2:** Thank you for this thoughtful question regarding the relationship between our method and importance sampling or domain adaptation weighting schemes.
> - Our method differs from standard domain adaptation in two key aspects:
>     1. most domain adaptation methods assume a different distribution of $X$ but consistent distribution of $Y\mid X$ across domains (covariate shift setting)
>     2. in the setting of our paper, we assume arbitrary distribution shifts between auxiliary and target data (including both covariate and label shifts), due to the corrupted nature of the auxiliary data. This makes traditional covariate weighting schemes unsuitable for loss estimation in our setting.
> - On the other hand, while the estimation of $\phi_2$ indeed incorporates importance sampling principles, our methodology extends beyond conventional implementations. Specifically, we utilize the auxiliary data's joint distribution over both covariates and labels as our proposal distribution for target loss estimation, rather than relying solely on marginal covariate distributions.
>
>
> **W3\&Q3:** Thank you for raising this important concern about potential bias in the labeled target set $L_t$. This is an insightful observation about our density ratio estimation procedure.
> - First, your point is particularly relevant, and we would like to clarify that our method's design naturally accommodates this form of bias. While active learning does indeed induce a shift in the covariate distribution of $L_t$ (as certain examples are preferentially selected), the conditional distribution of $Y\mid X$ remains unaffected since we always observe the true labels for selected points.
> - Crucially, as shown in Eq. (1) of our origin paper and the derivation between lines 127-128, our estimation of $\phi_2$ depends only on the ratio $r(Y\mid x)$ of conditional distributions rather than the marginal covariate distributions. This means that while the composition of $L_t$ may be biased in terms of $x$, this bias does not compromise the validity of our density ratio estimates for the purpose of loss reweighting, since the key quantity $r(y\mid x)$ can still be accurately estimated from the available data.
>
>
> **W4:** Thank you for raising this important point about model dependency in our auxiliary uncertainty estimation.
> - We acknowledge that our method is model-dependent and that errors in the trained auxiliary model will propagate into the selection process. However, we emphasize that these estimation errors are correlated with the true loss, which provides meaningful signal for sample selection.
> - Our empirical results demonstrate that simple models like random forests can provide meaningful uncertainty estimates for effective sample selection. Crucially, when using appropriate auxiliary data, the selection performance with these imperfect estimates still outperforms the BMDAL baseline that ignores this term (by setting it to 1).
>
> **W5:** Thank you for raising these important concerns regarding computational complexity and scalability.
> - Our method introduces additional components (density ratio estimation and auxiliary loss estimation) compared to gradient-only approaches BMDAL. However the selection kernel itself remains unchanged; we only modify the features used for selection. The primary computational overhead stems from processing auxiliary data. For efficiency, lightweight machine learning methods (e.g., random forests) can be adopted for density ratio and loss estimation without compromising performance.
> - To illustrate scalability, we conducted experiments across 5 real-world datasets with varying auxiliary dataset sizes (100, 500, 1000, 10,000 samples) on a server with dual Intel Xeon Gold 6330 CPUs (112 threads total) and 125GB RAM.
> - The results are shown in the table below. Even with only 100 auxiliary data, the maxdet selection strategy, which performs best among the compared approaches, still achieves meaningful performance improvements. The gain grows with larger auxiliary datasets. Crucially, for auxiliary datasets below 1,000 samples, the additional time and memory overhead remains negligible.
> - In practical applications, we can choose the auxiliary data size based on their specific needs for either better performance (larger auxiliary sets) or faster computation (smaller auxiliary sets).
>
> We'll add these discussions in the revised version.
>
> |||lcmd|maxdiag|maxdet|fw|maxdist|bait|kmeanspp|
> |-|-|-|-|-|-|-|-|-|
> |**AvgUsg(M)**|BMDAL|571.40|502.20|553.80|544.00|576.00|596.60|562.20|
> ||ours(100)|575.20|501.40|561.40|570.60|570.00|599.00|511.00|
> ||ours(500)|593.40|522.00|578.60|567.00|587.00|609.00|520.20|
> ||ours(1000)|595.00|529.00|584.80|578.00|585.80|610.80|539.20|
> ||ours(10000)|766.00|713.40|750.00|752.40|770.60|769.20|765.40|
> |**AvgTime(s)**|BMDAL|1.39|0.06|0.32|0.23|1.19|1.10|1.28|
> ||ours(100)|2.35|0.91|1.23|1.12|2.10|2.02|2.41|
> ||ours(500)|3.08|1.62|1.97|1.83|2.80|2.72|2.81|
> ||ours(1000)|3.57|2.27|2.62|2.46|3.39|3.32|3.17|
> ||ours(10000)|15.04|14.49|15.23|14.74|15.95|15.44|15.89|
> |**AvgImpro(%)**|ours(100)|-11.47|14.91|**5.83**|2.28|-9.17|5.86|-0.66|
> ||ours(500)|-6.17|20.31|**10.69**|4.66|-0.40|8.95|2.60|
> ||ours(1000)|-5.65|17.99|**10.47**|5.03|1.13|7.71|3.86|
> ||ours(10000)|-6.08|22.56|**12.02**|5.20|1.91|8.76|4.47|
>
> **W6:** Thank you for your valuable feedback regarding the readability of Section 2.3. We apologize for the current mathematical density in this section and fully agree that additional intuitive explanations would enhance understanding. In our revision we will add more intuitive struture about the NTK-based decomposition.
>
> **Q5:** Our approach can be extended to classification tasks by using appropriate loss functions such as cross-entropy. However, we note an important technical distinction: since class labels are discrete rather than continuous, the joint density ratio estimation would decompose into estimating separate covariate density ratios within each class-specific subset of the auxiliary data.
>
> *We sincerely appreciate your thoughtful and constructive feedback. Many of the points you raised have helped us identify important directions for future research. We are more than happy to answer any further questions during the rebuttal period.*

---

### Decision · Program_Chairs · 2025-09-17

**Decision:**

Accept (poster)

**Comment:**

This paper introduces AGBAL, an active learning framework for regression that leverages auxiliary data to improve uncertainty estimation. By applying density ratio weighting to recalibrate auxiliary losses, AGBAL enables more reliable sample selection even under distribution shifts. Theoretical analysis supports the approach, and experiments on synthetic and real-world datasets demonstrate consistent improvements over methods that do not use auxiliary data.

Reviewers recognized the novelty of the approach and its effectiveness, validated through both theoretical analysis and experimental results. The authors’ rebuttal clarified several concerns raised in the original reviews. As a result, all reviewers supported acceptance. The authors are encouraged to incorporate the additional discussions and results into the final version.